

# An advanced method of contributing emissions to short-lived chemical species (OH and HO$_2$): The TAGGING 1.1 submodel based on the Modular Earth Submodel System (MESSy 2.53)

Vanessa S. Rieger[1,2], Mariano Mertens[1], and Volker Grewe[1,2]

[1]Deutsches Zentrum für Luft- und Raumfahrt, Institut für Physik der Atmosphäre, Oberpfaffenhofen, Germany
[2]also at: Delft University of Technology, Aerospace Engineering, Section Aircraft Noise and Climate Effects, Delft, Netherlands

*Correspondence to:* Vanessa S. Rieger (vanessa.rieger@dlr.de)

**Abstract.** To mitigate human impact on climate change, it is essential to determine the contribution of emissions to the concentration of certain trace gases. This study presents an advanced version of the tagging method for OH and HO$_2$ (HO$_x$) which attributes HO$_x$ concentration to emissions. While the former version V1.0 only considered 12 reactions in the troposphere, the new version V1.1, presented here, takes 19 reactions in the troposphere into account. For the first time, also the main chemical reactions for the HO$_x$ chemistry in the stratosphere are regarded (in total 27 reactions). To fully take into account the main HO$_2$ source by the reaction of H and O$_2$, the tagging of H radical is introduced. In order to close the budget between the sum of all contributions and the total concentration, we explicitly introduce rest terms, which balance the deviation of HO$_x$ production and loss. The contributions to the OH and HO$_2$ concentration obtained by the improved tagging method V1.1 deviates from V1.0 in certain source categories. For OH, major changes are found in the categories of biomass burning emissions, biogenic emissions and methane decomposition. For HO$_2$, the contributions differs strongly in the categories biogenic emission and methane decomposition. The tagged long-lived species of reactive nitrogen compounds NO$_y$, non-methane hydrocarbons NMHC and peroxyacyl nitrates PAN show only little changes. O$_3$ from biogenic emissions and methane decomposition decreases in the tropical troposphere. Variations for CO from lightning, biogenic and methane decomposition are found in the Southern Hemisphere.

## 1 Introduction

The radicals hydroxyl OH and hydroperoxyl HO$_2$ are crucial for the atmospheric chemistry. Both radicals are very reactive and have a lifetime of only a few seconds. OH is frequently converted to HO$_2$ and vice versa. Thus, the OH and HO$_2$ radicals are closely linked and often referred together as the chemical family HO$_x$. The ratio of OH to HO$_2$ in an air parcel strongly depends on the chemical background, in particular on the composition of nitrogen oxides NO$_x$ (= NO + NO$_2$) and non-methane hydrocarbons NMHC (Heard and Pilling, 2003).

HO$_x$ impacts the global warming and the local air quality in various ways: Reacting with greenhouse gases such as methane CH$_4$ and ozone O$_3$, OH reduces their atmospheric residence time (e.g. Stevenson et al. (2006); Voulgarakis et al. (2013);




Righi et al. (2015)). Hence, $HO_x$ controls the impact of $CH_4$ and $O_3$ on global warming. Moreover, being the main oxidizer in the troposphere, OH is involved in the decomposition of pollutants, the production of ground-level ozone, photochemical smog and secondary organic aerosols (e.g. Lawrence et al. (2001); Heard and Pilling (2003)). Consequently, to quantify the human impact on the climate and air quality, it is essential to understand the distribution and variability of OH and $HO_2$ in the
atmosphere.

However, the determination of OH and $HO_2$ concentrations in the atmosphere is still challenging due to their short lifetimes. In field campaigns $HO_x$ concentrations are measured on a local scale which is generally difficult to compare with global models (e.g. Ren et al. (2003); Olson et al. (2006)). For certain environments, model studies compare well with the measurements. Other regions, such as unpolluted forest areas, show large discrepancies (Heard and Pilling, 2003; Stone et al., 2012). On a
regional and global scale, no direct $HO_x$ measurement is available. So far, OH can only be estimated indirectly by measurements and emission rates of methyl chloroform $CH_3CCl_3$ (Prinn et al., 2005). As emissions of $CH_3CCl_3$ steadily decline, Liang et al. (in press, 2017) suggest an alternative method: They combine several trace gases such as $CH_2F_2$, $CH_2FCF_3$, $CH_3CHF_2$ and $CHClF_2$ in a gradient-trend based two-box model approach to derive a global OH concentration of $11.2 \cdot 10^5$ molec $cm^{-3}$. Overall, global climate-chemistry models estimate a tropospheric OH concentration of around $11 \cdot 10^5$ molec $cm^{-3}$ (Naik
et al., 2013), which compares well with the observation-based results from Prinn et al. (2005) and Liang et al. (2017).

To mitigate the human impact on climate change, it is crucial to determine the contribution of a specific emission sector to the concentration of certain chemical species (Grewe et al., 2012). To do so, we use the so called "tagging" method (Grewe et al., 2010, 2017). This method splits up all chemical species which are important for $O_3$ production and destruction into ten source categories: emissions from anthropogenic non-traffic (e.g. industry and households), road traffic, ship traffic, air traffic,
biogenic sources, biomass burning, lightning, methane and nitrous oxide decompositions and stratospheric ozone production. Subsequently, the contributions of theses sources to the concentrations of $O_3$, CO, OH, $HO_2$, peroxyacyl nitrates PAN, reactive nitrogen compounds $NO_y$ (NO, $NO_2$, $HNO_4$, ...) and non-methane hydrocarbons NMHC are diagnosed. The contribution calculations are based on chemical reaction rates, online emissions (e.g. lightning), offline emissions (e.g. road traffic) and depositions rates. It considers the competition of $NO_y$, CO and NMHC in producing and destroying $O_3$.

The long-lived species $O_3$, CO, PAN, $NO_y$ and NMHC are tagged in a different way than the short-lived species OH and $HO_2$. (In this study, $O_3$, CO, PAN, $NO_y$ and NMHC are denoted as long-lived species because their atmospheric lifetime is significantly longer then the lifetime of OH and $HO_2$.) For these long-lived species, each source specific tracer is transported, receives the corresponding online or offline emissions, is deposited and reacts with other species. Based on these processes, the tagging method determines the concentration of the source specific tracers.

However, the short-lived species $HO_x$ are not transported and experience neither emissions nor deposition. Thus, the tagging method for long-lived species is not applicable. Tsati (2014) and Grewe et al. (2017) introduced a modified tagging method for $HO_x$: since the lifetime of OH and $HO_2$ is very short, a steady-state between the production and destruction of OH and $HO_2$ is assumed. Using the main chemical reactions of the $HO_x$ chemistry, the contribution of each source category to OH and $HO_2$ can be determined.





But the $HO_x$ tagging method V1.0, presented by Grewe et al. (2017), did not consider all relevant reactions for the production and loss of $HO_x$. Especially, the reactions which are important in the stratosphere were not taken into account. Moreover, the budget of the sum of all tagged $HO_x$ species and the total $HO_x$ concentrations was not closed. In this study, we present a revised version V1.1 of the $HO_x$ tagging method, largely improving these shortcomings. It includes the main chemical reactions of

the $HO_x$ chemistry in the troposphere and stratosphere. This is enabled by introducing the tagging of the hydrogen radical H. Special care is taken for the closure of the budget.

The paper is structured as follows: After introducing the model setup in Section 2, we present the advanced $HO_x$ tagging mechanism V1.1 in Section 3. In Section 4, the results are compared with the tagging mechanism V1.0 by Grewe et al. (2017). Finally, Section 5 concludes the method and the results of this study.

**2   Model description of EMAC and MECO(n)**

To evaluate the further developed $HO_x$ tagging mechanism we use the same model setup as Grewe et al. (2017). A global climate simulation is performed with the ECHAM/MESSy Atmospheric Chemistry (EMAC) climate-chemistry model. EMAC is a numerical chemistry and climate simulation system that includes submodels describing tropospheric and middle atmosphere processes and their interaction with oceans, land and human influences (Jöckel et al., 2010). It uses the second version of the

Modular Earth Submodel System (MESSy2) to link multi-institutional computer codes. The core atmospheric model is the 5th generation European Centre Hamburg general circulation model (ECHAM5, Roeckner et al. (2006)). For the present study we apply EMAC in the T42L90MA-resolution, i.e. with a spherical truncation of T42 (corresponding to a quadratic Gaussian grid of approx. 2.8° by 2.8° in latitude and longitude) with 90 vertical hybrid pressure levels up to 0.01 hPa. For the simulation presented in this study, the time span of July 2009 to December 2010 is considered: half a year as a spin-up and one year for

the analysis.

For the chemical scheme, we use the submodel MECCA (Module Efficiently Calculating the Chemistry of the Atmosphere) which is based on Sander et al. (2011) and Jöckel et al. (2010). The chemical mechanism includes 218 gas phase, 12 heterogeneous and 68 photolysis reactions. In total 188 species are considered. It regards the basic chemistry of OH, $HO_2$, $O_3$, $CH_4$, nitrogen oxides, alkanes, alkenes, chlorine and bromine. Alkynes, aromatics and mercury are not considered.

Total global emissions of lightning $NO_x$ are scaled to approximately 4 Tg(N) $a^{-1}$ (parametrized after Grewe et al. (2001)). The submodel ONEMIS (Kerkweg et al., 2006) calculates $NO_x$ emissions from soil (parametrized after Yienger and Levy (1995)) and biogenic $C_5H_8$ emissions (parametrized after Guenther et al. (1995)). Direct $CH_4$ emissions are not considered, instead pseudo-emissions are calculated using the submodel TNUDGE (Kerkweg et al., 2006). This submodel relaxes the mixing ratios in the lowest model layer towards observations by Newtonian relaxation (more details are given by Jöckel et al.

30  (2016)).

To show the effect of the $HO_x$ tagging method on a regional scale, a further simulation with the coupled model system MESSyfied ECHAM and COSMO models nested n-times (MECO(n)) is performed. The nested system couples the global chemistry-climate model EMAC online with the regional chemistry climate model COSMO/MESSy (Kerkweg and Jöckel,





2012a, b). To test the HO$_x$ tagging in MECO(n), we conduct a simulation using one COSMO/MESSy nest over Europe with a resolution of 0.44°. EMAC is applied in a horizontal resolution of T42 with 31 vertical levels. The period from July 2007 to December 2008 is simulated. The setup of the simulation is identical to the one described in Grewe et al. (2017). A detailed chemical evaluation of the setup is given in Mertens et al. (2016).

Both model simulations are based on the quasi chemistry-transport model (QCTM) mode in which the chemistry is decoupled from the dynamics (Deckert et al., 2011). The anthropogenic emissions are taken from the MACCity emission cataster (Granier et al., 2011). The TAGGING submodel (as described by Grewe et al. (2017)) calculates the contributions of source categories to O$_3$, CO, NO$_y$, PAN and NMHC concentration. The contributions of OH and HO$_2$ are calculated with the advanced method V1.1 presented in the next section.

## 3   Tagging method of short-lived species

### 3.1   Tagging method V1.0

The tagging mechanism V1.0 described by Grewe et al. (2017) determines the contribution of source categories to O$_3$, NO$_y$, CO, NMHC, PAN, OH and HO$_2$ concentrations. Ten source categories are considered: For example, the concentration of O$_3$ is split up into O$_3$ from anthropogenic non-traffic (e.g. industry) emissions O$_3^{ant}$, road traffic emissions O$_3^{tra}$, ship emissions O$_3^{shp}$,

air traffic emissions O$_3^{air}$, biogenic emissions O$_3^{bio}$, biomass burning O$_3^{bb}$, lightning O$_3^{lig}$, methane decomposition O$_3^{CH_4}$, nitrous oxide decomposition O$_3^{N_2O}$ and stratospheric ozone production O$_3^{str}$. These tagged species go through the same chemical reactions and the same deposition loss processes as O$_3$. The tagging method uses a combinatoric approach to determine the contributions: It redistributes the production and loss rates of each species to the ten source categories according to the concentrations of the tagged species. Details on the tagging theory and implementation in EMAC and MECO(n) are found in

Grewe (2013) and Grewe et al. (2017), respectively.

For the first time, V1.0 determined the contribution of source categories to OH and HO$_2$ concentrations. The mechanism V1.0 was based on 12 reactions for the HO$_x$ chemistry (reactions marked with "o" in last column of Table 1). It included the main production and loss reactions of HO$_x$ with O$_3$, NO$_y$, NMHC, CO and CH$_4$. V1.0 only regarded reactions which are important in the troposphere. Reactions which mainly occur in the stratosphere were not taken into account. However, the main

HO$_2$ production by the reaction (1) H + O$_2$ ⟶ HO$_2$ was not explicitly regarded. It was combined with reaction (11) CO + OH ⟶ H + CO$_2$ (see Table 1) to

$$CO + OH \longrightarrow CO_2 + H \xrightarrow{O_2} CO_2 + HO_2 \tag{1}$$
$$\Longrightarrow CO + OH \xrightarrow{O_2} CO_2 + HO_2 \tag{2}$$

But not all H radicals in the troposphere are produced by the reaction of CO + OH. Also the reactions (7) OH + O($^3$P), (10) H$_2$

+ OH and (31) photolyses of formaldehyde HCHO produce H (Table 3). These reactions were neglected in V1.0. Thus, only 80 % of the H production and therefore only 80 % of the HO$_2$ production by reaction (1) was considered. In the stratosphere,

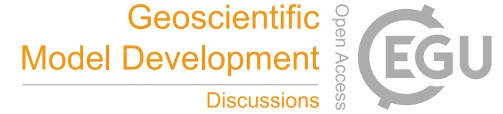



the reaction of CO + OH becomes less important and most of H is produced by reactions (7) and (11). Consequently, only 6 % of the H and also of $HO_2$ production by reaction (1) was regarded by this approach.

In the troposphere, the most important reactions not covered in V1.0 are reaction (1) H + $O_2$ as well as reaction (15) $NO_2$ + $HO_2$ and (18) decomposition of $HNO_4$. In the stratosphere, reactions (1) H + $O_2$, (5) $HO_2$ + $O(^3P)$ and (7) OH + $O(^3P)$ play a

leading role and were not included in V1.0.

Most reaction rates were obtained directly by the MECCA mechanism of EMAC. Each reaction occurring in a simulation was precisely added up. However for reactions with NMHC, the rates were obtained indirectly. The reaction rate of OH with NMHC (reaction 22) was determined via the production of CO by assuming that each reaction produces one CO molecule. For the reaction rates of NO and $HO_2$ with NMHC, only the reaction with methylperoxy radical $CH_3O_2$ was considered.

For the OH and $HO_2$ production and loss, a steady state was assumed: the production equaled the loss. Based on this assumption, the source contributions to the concentrations of OH and $HO_2$ were determined. However, the sum over the contributions of all ten source categories to the OH and $HO_2$ concentrations did not equal the total OH and $HO_2$ concentrations. It deviated by about 70 %. To close the budget and to extend to method to the stratosphere, an advanced $HO_x$ tagging method V1.1 is developed in this study.

**3.2 Reduced $HO_x$ reaction system V1.1**

OH and $HO_2$ react with many chemical species. To reduce the calculation time of a simulation, we boil down the $HO_x$ chemistry to the most important reactions which occur in the troposphere and stratosphere. We consider only reactions with a rate larger than $10^{-15}\,\mathrm{mol\,mol^{-1}s^{-1}}$ (Reactions 1 to 30 in Table 1). Hence, we increase the number of reactions from 12 (V1.0) to 30 (V1.1), which still constitutes a reduced set of reactions compared to the chemical scheme used in EMAC. In the following,

we call this set *reduced $HO_x$ reaction system V1.1*.

The reactions which are important in the troposphere are explicitly indicated in Table 1. As stated above, the reaction (1) of H and $O_2$ dominates the $HO_2$ production in the troposphere. In the mechanism V1.0, only part of this $HO_2$ source was regarded (see Sect. 3.1). The most important $HO_2$ loss is the reaction with NO (reaction 14) followed by the reaction with itself producing $H_2O_2$ (reaction 3). The production via $H_2O$ and $O(^1D)$ produces most of the OH which can be found in the

troposphere (reaction 2). The excited oxygen molecule $O(^1D)$ originates from the photolysis of $O_3$. Also reaction (14) of NO and $HO_2$ produces a major part of tropospheric OH. OH is mostly destroyed by CO (reaction 11) followed by the destruction by NMHC (reaction 22).

In the stratosphere different chemical reactions become important. Here, OH is mainly destroyed by $O_3$, producing most of the $HO_2$ in the stratosphere. The reaction is partly counteracted by the reaction (14) which produces OH and destroys $HO_2$.

Since large quantities of $O_3$ can be found in the stratosphere, $O_3$ or the excited oxygen radical $O(^3P)$ destroys $HO_2$. Reactions with NMHC, CO and $CH_4$ play only a minor role in the stratosphere.

The reactions of OH and $HO_2$ with chlorine and bromide were not considered in V1.0. We add these reactions, which occur only in the stratosphere, to the tagging mechanism V1.1. The photolysis of $H_2O_2$ (reaction 19), HOCl (reaction 28) and HOBr



| | reaction | | | rates | tropos. | stratos. | V1.1 |
|---|---|---|---|---|---|---|---|
| 1 | H + O$_2$ | $\longrightarrow$ | HO$_2$ | $R_1$ | x | x | (x) |
| 2 | H$_2$O + O($^1$D) | $\longrightarrow$ | 2 OH | $R_2$ | x | x | o |
| 3 | HO$_2$ + HO$_2$ | $\longrightarrow$ | H$_2$O$_2$ + O$_2$ | $R_3$ | x | | o |
| 4 | HO$_2$ + O$_3$ | $\longrightarrow$ | OH + 2 O$_2$ | $R_4$ | x | x | o |
| 5 | HO$_2$ + O($^3$P) | $\longrightarrow$ | OH + O$_2$ | $R_5$ | | x | x |
| 6 | OH + O$_3$ | $\longrightarrow$ | HO$_2$ + O$_2$ | $R_6$ | x | x | o |
| 7 | OH + O($^3$P) | $\longrightarrow$ | H + O$_2$ | $R_7$ | | x | x |
| 8 | HO$_2$ + OH | $\longrightarrow$ | H$_2$O + O$_2$ | $R_8$ | x | x | o |
| 9 | H$_2$O$_2$ + OH | $\longrightarrow$ | H$_2$O + HO$_2$ | $R_9$ | x | | x |
| 10 | H$_2$ + OH | $\longrightarrow$ | H$_2$O + H | $R_{10}$ | x | | x |
| 11 | CO + OH | $\longrightarrow$ | H + CO$_2$ | $R_{11}$ | x | x | o |
| 12 | CH$_4$ + OH | $\longrightarrow$ | CH$_3$ + H$_2$O | $R_{12}$ | x | x | o |
| 13 | ClO + OH | $\longrightarrow$ | 0.94 Cl + 0.94 HO$_2$ + 0.06 HCl + 0.06 O$_2$ | $R_{13}$ | | x | x |
| 14 | NO + HO$_2$ | $\longrightarrow$ | NO$_2$ + OH | $R_{14}$ | x | x | o |
| 15 | NO$_2$ + HO$_2$ | $\longrightarrow$ | HNO$_4$ | $R_{15}$ | x | x | x |
| 16 | NO + OH | $\longrightarrow$ | HONO | $R_{16}$ | | x | x |
| 17 | NO$_2$ + OH | $\longrightarrow$ | HNO$_3$ | $R_{17}$ | | x | o |
| 18 | HNO$_4$ | $\longrightarrow$ | NO$_2$ + HO$_2$ | $R_{18}$ | x | | x |
| 19 | H$_2$O$_2$ + $h\nu$ | $\longrightarrow$ | 2 OH | $R_{19}$ | x | | - |
| 20 | HONO + $h\nu$ | $\longrightarrow$ | NO + OH | $R_{20}$ | | x | x |
| 21 | HNO$_3$ + $h\nu$ | $\longrightarrow$ | NO$_2$ + OH | $R_{21}$ | | x | x |
| 22 | NMHC + OH | $\longrightarrow$ | NMHC | $R_{22}$ | x | | o |
| 23 | NMHC + HO$_2$ | $\longrightarrow$ | NMHC | $R_{23}$ | x | | o |
| 24 | NMHC + NO$_y$ | $\longrightarrow$ | HO$_2$ + NMHC + NO$_y$ | $R_{24}$ | x | x | o |
| 25 | NMHC + OH | $\longrightarrow$ | NMHC + HO$_2$ | $R_{25}$ | x | | x |
| 26 | NMHC + $h\nu$ | $\longrightarrow$ | NMHC + HO$_2$ | $R_{26}$ | x | | x |
| 27 | ClO + HO$_2$ | $\longrightarrow$ | HOCl + O$_2$ | $R_{27}$ | | x | x |
| 28 | HOCl + $h\nu$ | $\longrightarrow$ | OH + Cl | $R_{28}$ | | x | - |
| 29 | BrO + HO$_2$ | $\longrightarrow$ | HOBr + O$_2$ | $R_{29}$ | | x | x |
| 30 | HOBr + $h\nu$ | $\longrightarrow$ | OH + Br | $R_{30}$ | | x | - |

**Table 1.** Reduced HO$_x$ reaction system V1.1. Main reactions which describe the HO$_x$ chemistry in troposphere and stratosphere. In the column "tropos." ("stratosp.") the reaction which are important in the troposphere (stratosphere) are marked. In the column "V1.1", reactions marked with "o" were already included in V1.0. Reactions marked with "x" are added in V1.1. Reactions marked with "(x)" were only partly taken into account in V1.0. Reactions marked with "-" are excluded as it is not possible to apply the tagging theory.



| | | OH | | HO$_2$ | |
|---|---|---|---|---|---|
| | | production | loss | production | loss |
| total - MECCA | tropos. | $5.11 \cdot 10^{-14}$ | $5.11 \cdot 10^{-14}$ | $5.13 \cdot 10^{-14}$ | $5.12 \cdot 10^{-14}$ |
| | *stratos.* | *$2.78 \cdot 10^{-13}$* | *$2.78 \cdot 10^{-13}$* | *$2.48 \cdot 10^{-13}$* | *$2.48 \cdot 10^{-13}$* |
| reduced - V1.1 all | tropos. | $5.01 \cdot 10^{-14}$ | $5.07 \cdot 10^{-14}$ | $4.95 \cdot 10^{-14}$ | $5.11 \cdot 10^{-14}$ |
| | *stratos.* | *$2.54 \cdot 10^{-13}$* | *$2.76 \cdot 10^{-13}$* | *$2.47 \cdot 10^{-13}$* | *$2.48 \cdot 10^{-13}$* |
| reduced - V1.1 tag | tropos. | $4.58 \cdot 10^{-14}$ | | | |
| | *stratos.* | *$2.50 \cdot 10^{-13}$* | | | |
| reduced - V1.0 | tropos. | $4.54 \cdot 10^{-14}$ | $4.95 \cdot 10^{-14}$ | $3.10 \cdot 10^{-14}$ | $4.38 \cdot 10^{-14}$ |
| | *stratos.* | *$8.60 \cdot 10^{-14}$* | *$1.30 \cdot 10^{-13}$* | *$1.19 \cdot 10^{-13}$* | *$8.30 \cdot 10^{-14}$* |

**Table 2.** Annual mean of OH and HO$_2$ production and loss rates in mol mol$^{-1}$ s$^{-1}$ for the total HO$_x$ production and loss (derived from the MECCA scheme in EMAC) and for the production and loss of the reduced reaction system of the tagging mechanism V1.0 and V1.1. For OH, also the production rate for all reactions listed in Table 1 (indicated with "all") and these which are finally used in the tagging mechanism (without reaction (19), (28) and (30), indicated with "tag") are shown. The first row gives the rates for the troposphere, the second row for the stratosphere (written in italic).

(reaction 30) are not considered in the HO$_x$ tagging mechanism V1.1 because they do not fulfil the steady-state assumption (see Sect. 3.4 for more details).

### 3.3 Steady-state assumption

To correctly describe the HO$_x$ chemistry, it is crucial that HO$_x$ production and loss of the reduced HO$_x$ reaction system V1.1 in

Table 1 (almost) equal the HO$_x$ production and loss of the complete HO$_x$ chemistry. Fig. 1 compares the seasonal cycle (year 2010) of the HO$_x$ production and loss of the reduced reaction system for the tagging mechanisms V1.0 and V1.1 and the total production and loss rates derived from the MECCA mechanism in EMAC. The production and loss rates are obtained from an EMAC simulation following the setup described in Sect. 2. Furthermore, Fig. 1 shows the OH production rate as a sum of all 30 reactions (indicated with "all") and as a sum of the 27 reactions (without reaction 19, 28 and 30) which are finally used

in the tagging mechanism (indicated with "tag"). The annual mean values are further summarized in Table 2. In this section, we only discuss the OH production rate of all 30 reactions ("all"). The implications of the OH production rate "tag" with 27 reactions are discussed in Sect. 3.4.

In general, the total OH production (derived by MECCA) equals the total OH loss in the troposphere and stratosphere. The same holds for HO$_2$. For this reason, the grey line for the total loss rate in Fig. 1 is not visible since it overlaps with the total

production rate (black line).

In the troposphere, the yearly mean of the total OH production and loss rates are $5.11 \cdot 10^{-14}$ mol mol$^{-1}$ s$^{-1}$ (Fig. 1a, Table 2). The OH loss of the reduced HO$_x$ reaction system V1.1 represents almost the total OH loss occurring in the troposphere.

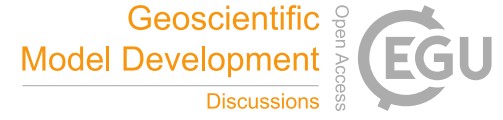



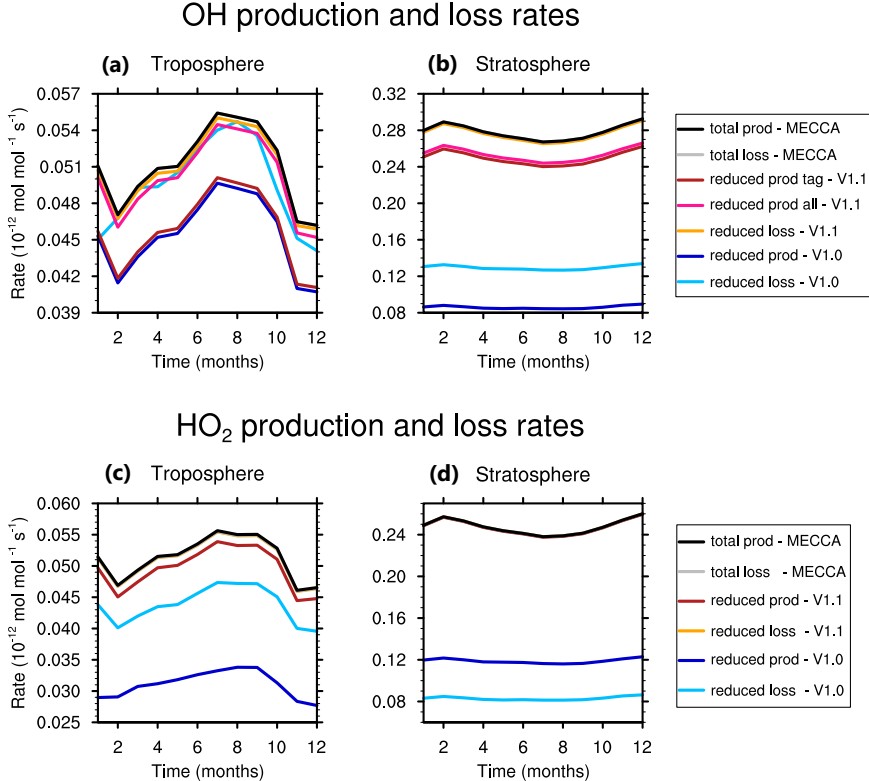

**Figure 1.** Monthly mean of OH and HO$_2$ production and loss rates for the total HO$_x$ production and loss (derived from the MECCA scheme in EMAC) and for the production and loss of the reduced reaction system of the tagging mechanism V1.0 and V1.1. (a) and (c) shows the rates for troposphere, (b) and (d) for the stratosphere. For OH, also the production rate for all reactions listed in Table 1 (indicated with "all") and these which are finally used in the tagging mechanism (without reaction (19), (28) and (30), indicated with "tag") are shown.

The yearly averaged OH loss of V1.1 deviates only by less than 1 % from the total OH loss. In contrast, the OH loss of V1.0 deviated by more than 3 % from the total OH loss. The OH production for V1.1 ("all") differs by 2 % from the total OH production. This is clearly less than for V1.0 which differed by 11 % from the total OH production.

Considering HO$_2$ in the troposphere, the total loss rate of $5.12 \cdot 10^{-14}$ mol mol$^{-1}$ s$^{-1}$ is very well reflected by the reduced
5 HO$_x$ reaction system V1.1. It deviates only 0.1 % from the total HO$_2$ loss (thus the orange line is not visible in Fig. 1c). In contrast, the HO$_2$ loss of V1.0 differed by 15 %. The HO$_2$ production of V1.1 disagrees by 3 %, V1.0 by 40 % from the total production.

In the stratosphere, the total OH production of $2.78 \cdot 10^{-13}$ mol mol$^{-1}$ s$^{-1}$ equals the total OH loss. Analogously , HO$_2$ production of $2.48 \cdot 10^{-13}$ mol mol$^{-1}$ s$^{-1}$ balances the total loss. Since V1.0 was only developed for the troposphere, not
10 all reactions which are important in the stratosphere were considered. Thus, the OH and HO$_2$ production and loss rates of V1.0 considerably underestimated the total production and loss rates. The OH production of V1.1 misses 9 % of the total OH




production in the stratosphere. However, the OH loss is presented very well with an error of only 0.7 %. The $HO_2$ loss in the stratosphere is perfectly presented by V1.1. The $HO_2$ production of V1.1 deviates by only 0.2 % from the total $HO_2$ production.

All reaction rates, except OH loss and $HO_2$ production of V1.0, follow a similar seasonal cycle. In V1.0 the rate of reaction (18) and (22) was determined indirectly by the increase and decrease of CO (see Sect. 3.1). It was assumed that for each NMHC

which is destroyed by OH a CO molecule is produced. In V1.1 the reaction rate is determined directly by precisely counting each single reaction. Thus, OH loss and $HO_2$ production of V1.1 follow exactly the trend of the total production and loss rates over the year. The deviations between the total production and loss rates and those of V1.1 stay constant over time.

The reduced $HO_x$ reaction system V1.1 represents very well the total $HO_x$ production and loss in the troposphere and stratosphere. The deviations between the $HO_x$ production and loss of the reduced reaction system and the complete $HO_x$

chemistry are small. Thus, the steady-state assumption for OH and $HO_2$ of the reduced $HO_x$ reaction system V1.1 is justified.

## 3.4   Deductions of tagged species

To derive how much OH and $HO_2$ is produced and destroyed by a source category $i$, the tagging approach described in Grewe et al. (2010, 2017) is used. In general, bimolecular reactions with two chemical species A + B $\longrightarrow$ C are tagged as follows: Each tagged species is split up into its contribution from $n$ source categories $A = \sum_{i=1}^{n} A^i$, $B = \sum_{i=1}^{n} B^i$ and $C = \sum_{i=1}^{n} C^i$.

These subspecies $(A^i, B^i, C^i)$ go through the same reactions as their main species $(A, B, C)$. If $A$ from category $i$ reacts with $B$ from category $j$, then the resulting species $C$ belongs half to the category $i$ and half to the category $j$:

$$A^i + B^j \longrightarrow \frac{1}{2}C^i + \frac{1}{2}C^j \tag{3}$$

Consequently, the production $P$ and loss $L$ of a species from the category $i$ (here $LossA^i$, $LossB^i$ and $ProdC^i$) can be determined by regarding all possible combinations of the reaction between $A^i$ and $B^j$:

$$LossA^i = LossB^i = ProdC^i = k\left(A^iB^i + \sum_{j=1, j\neq i}^{n} \frac{1}{2}A^iB^j + \sum_{j=1, j\neq i}^{n} \frac{1}{2}A^jB^i\right) = \frac{1}{2}R\left(\frac{A^i}{A} + \frac{B^i}{B}\right) \tag{4}$$

with $k$ being the reaction rate coefficient and $R = kAB$ being the respective reaction rate. For unimolecular reactions A $\longrightarrow$ B + C, the distribution of categories from the educts is completely passed to the products:

$$LossA^i = LossB^i = ProdC^i = R\frac{A^i}{A} \tag{5}$$

As described above, the long-lived species $O_3$, CO, $NO_y$ and NMHC are tagged explicitly. However, other species such as

H, $H_2$, $H_2O_2$, $CH_4$, ClO, BrO, HOCl, HOBr are not tagged (as in V1.0). Here, we need to apply different methods to retain the ratio of contribution to total concentration $\frac{A^i}{A}$:

1. If a tagged species reacts with a non-tagged species, the non-tagged species does not contribute and the tagging method for a unimolecular reaction is applied. Examples are reactions (9), (10) and (14).





| | reaction | | | rates | tropos. | stratos. |
|---|---|---|---|---|---|---|
| 1 | $H + O_2$ | $\longrightarrow$ | $HO_2$ | $R_1$ | x | x |
| 7 | $OH + O(^3P)$ | $\longrightarrow$ | $H + O_2$ | $R_7$ | | x |
| 10 | $H_2 + OH$ | $\longrightarrow$ | $H_2O + H$ | $R_{10}$ | x | |
| 11 | $CO + OH$ | $\longrightarrow$ | $H + CO_2$ | $R_{11}$ | x | x |
| 31 | $HCHO + h\nu$ | $\longrightarrow$ | $H + CO + HO_2$ | $R_{31}$ | x | |

**Table 3.** Reduced H reaction system showing the main reactions of H. In the column "tropos." ("stratosp.") the reaction which are important in the troposphere (stratosphere) are marked.

2. Using the family concept as described in Grewe et al. (2017) allows the assumption that all tags are distributed equally among the species in the same chemical family. It follows:

$$\frac{NO^i}{NO} = \frac{NO_2^i}{NO_2} = \frac{HNO_4^i}{HNO_4} = \frac{NO_y^i}{NO_y} \qquad (6)$$

As mentioned in Grewe et al. (2017), all species which are frequently converted back and forth to ozone are considered as an "ozone storage". These species together with $O_3$ are lumped into one chemical family "ozone". Both $O(^1D)$ and $O(^3P)$ belong to this chemical family. Hence, as in Grewe et al. (2017), we apply the family concept and set:

$$\frac{O(^1D)^i}{O(^1D)} = \frac{O(^3P)^i}{O(^3P)} = \frac{O_3^i}{O_3} \qquad (7)$$

3. The ratio of the contribution to the total concentration can be determined by introducing an explicit tagging mechanism. In reaction (1), neither H nor $O_2$ are tagged. To obtain the proper contribution, we set up a specific tagging of H itself. The H radical is very reactive, so H production balances H loss. Table 3 presents the main reactions for H. Based on Table 3, we can set up the H production $ProdH^i$ and H loss $LossH^i$ for the contribution of a specific source category $i$:

$$ProdH^i = R_1 \frac{H^i}{H} \qquad (8)$$

$$LossH^i = \frac{1}{2}R_7 \left( \frac{OH^i}{OH} + \frac{O_3^i}{O_3} \right) + R_{10}\frac{OH^i}{OH} + \frac{1}{2}R_{11} \left( \frac{CO^i}{CO} + \frac{OH^i}{OH} \right) + R_{31}\frac{NMHC^i}{NMHC} \qquad (9)$$

As mentioned above, the family concept also sets $\frac{HCHO^i}{HCHO} = \frac{NMHC^i}{NMHC}$. Since the steady-state assumption applies for H, the H production per source category $i$ $ProdH^i$ equals the loss $LossH^i$. After setting eq. 8 and 9 equal to each other, we obtain:

$$\frac{H^i}{H} = \frac{1}{2}\frac{R_7}{R_1} \left( \frac{OH^i}{OH} + \frac{O_3^i}{O_3} \right) + \frac{R_{10}}{R_1}\frac{OH^i}{OH} + \frac{1}{2}\frac{R_{11}}{R_1} \left( \frac{CO^i}{CO} + \frac{OH^i}{OH} \right) + \frac{R_{31}}{R_1}\frac{NMHC^i}{NMHC} \qquad (10)$$

4. To include the OH production by the photolysis of hydrogen peroxide $H_2O_2$ (reaction 19), we would need to tag $H_2O_2$. Since the production and the loss of $H_2O_2$ are not balanced, we can not assume a steady-state. Thus, a similar tagging





approach is not valid for $H_2O_2$. Consequently, we exclude the reaction (19) from the $HO_x$ tagging mechanism. This reaction contributes 8 % to the total OH production in the troposphere.

5. Hypochlorous acid HOCl and hypobromous acid HOBr are photolysed in the stratosphere and produce OH (reaction 28 and 30). However, HOCl and HOBr are not tagged explicitly. Although the steady-state assumption is globally valid, locally the production and loss of HOCl and HOBr are not balanced everywhere. In the stratosphere, for about 65 % of the model gridboxes the production deviates by more than 10 % from the loss of HOCl and HOBr. In particular, in the transition area between day and night in the polar region, the production deviates strongly from the loss. Also at night where the reactions mostly occur, the steady-state is not fulfilled everywhere. Moreover, since both species are not radicals, their lifetimes can not be assumed to be short. Hence, we can not apply the tagging mechanism, so we have to omit the reactions (28) and (30) for the OH production (see Table 1).

Dropping reaction (19), (28) and (30) leads to a lower OH production. The resulting OH production rate is also given in Fig. 1 and Table 2 ("tag"). Only 90 % of the total OH production is represented in the troposphere; 8 percent points less than all 30 reactions listed in Table 1 (V1.1 "all"). In the stratosphere, the difference is negligible: only 0.5 percent points of the total OH production are not regarded. To compensate this deviation from production and loss rate, we introduce a rest term in Sect. 3.5.

### 3.5 Closure of the budget

In V1.0, the budget of the tagged OH and $HO_2$ was not closed: the sum over the contributions from all source categories did not balance the total concentration. The averaged deviations for OH and $HO_2$ in troposphere were about 70 % of the total concentrations. Since the stratosphere was not considered in V1.0, the deviations were even larger (104 % for OH and 89 % for $HO_2$). The improved mechanism V1.1 results in a better closure, but the sum over all source categories still deviates in the troposphere by about 27 % (23 %) from the total OH ($HO_2$) concentration. In the stratosphere, the deviation is about 21 %. This is already a large improvement compared to V1.0, but still not satisfying. Although the tagging mechanism V1.1 includes many new reactions, omitting reaction (19), (28) and (30) still hampers the closure of the budget.

To close the budget, the steady-state assumption must apply. However, since we consider a reduced $HO_x$ and H chemistry mechanism (Table 1 and Table 3), the production of OH, $HO_2$ and H does not exactly equal the loss (see also Fig. 1 and Table 2). Thus, we introduce rest terms *resOH*, *resHO_2* and *resH* for OH, $HO_2$ and H to compensate for these deviations from steady-state. Each rest term is calculated by subtracting the production rate of the reduced reaction system (Table 1 and 3) from the loss rate. Finally, the rest terms have to be divided by the number of source categories *n* to account for the contribution per source category.

Considering the rest terms *resOH*, *resHO_2* and *resH*, the sum of OH and $HO_2$ now perfectly balances the total OH and $HO_2$ concentrations. The deviations are negligible (below $10^{-4}$ % for OH and below $10^{-3}$ % for $HO_2$). Consequently, including the rest terms to the tagging mechanism is mandatory to close the budget.



## 3.6  Determination of HO$_x$ contributions

Taking the above considerations into account, we finally derive the OH and HO$_2$ production and loss terms per source category $i$. In the reduced HO$_x$ reaction system V1.1 (Table 1), OH is produced by the reactions (2) H$_2$O + O($^1$D), (4) HO$_2$ + O$_3$, (5) HO$_2$ + O($^3$P), (14) NO + HO$_2$, (19) H$_2$O$_2$ + $hv$, (20) HONO + $hv$, (21) HNO$_3$ + $hv$, (28) HOCl + $hv$ and (30) HOBr + $hv$.

However, reactions (19), (28) and (30) are excluded as H$_2$O$_2$, HOCl and HOBr are not in steady-state (see Sect. 3.4) . Applying the partitioning described in Sect. 3.4, the OH production for a specific source category $i$ $ProdOH^i$ is determined as follows:

$$ProdOH^i = 2 \cdot R_2 \frac{O_3^i}{O_3} + \frac{1}{2} R_4 \left( \frac{HO_2^i}{HO_2} + \frac{O_3^i}{O_3} \right) + \frac{1}{2} R_5 \left( \frac{HO_2^i}{HO_2} + \frac{O_3^i}{O_3} \right) + \frac{1}{2} R_{14} \left( \frac{NO_y^i}{NO_y} + \frac{HO_2^i}{HO_2} \right)$$
$$+ R_{20} \frac{NO_y^i}{NO_y} + R_{21} \frac{NO_y^i}{NO_y} \tag{11}$$

OH is destroyed by the reactions (6) OH + O$_3$, (7) OH + O($^3$P), (8) HO$_2$ + OH, (9) H$_2$O$_2$ + OH, (10) H$_2$ + OH, (11) CO + OH, (12) CH$_4$ + OH, (13) ClO + OH, (16) NO + OH, (17) NO$_2$ + OH, (22) NMHC + OH and (25) NMHC + OH. The OH loss per

source category $i$ $LossOH^i$ is:

$$LossOH^i = \frac{1}{2} R_6 \left( \frac{OH^i}{OH} + \frac{O_3^i}{O_3} \right) + \frac{1}{2} R_7 \left( \frac{OH^i}{OH} + \frac{O_3^i}{O_3} \right) + \frac{1}{2} R_8 \left( \frac{HO_2^i}{HO_2} + \frac{OH^i}{OH} \right) + \frac{1}{2} R_9 \left( \frac{HO_2^i}{HO_2} + \frac{OH^i}{OH} \right)$$
$$+ R_{10} \frac{OH^i}{OH} + \frac{1}{2} R_{11} \left( \frac{CO^i}{CO} + \frac{OH^i}{OH} \right) + R_{12} \frac{OH^i}{OH} + R_{13} \frac{OH^i}{OH} + \frac{1}{2} R_{16} \left( \frac{NO_y^i}{NO_y} + \frac{OH^i}{OH} \right)$$
$$+ \frac{1}{2} R_{17} \left( \frac{NO_y^i}{NO_y} + \frac{OH^i}{OH} \right) + \frac{1}{2} R_{22} \left( \frac{NMHC^i}{NMHC} + \frac{OH^i}{OH} \right) + \frac{1}{2} R_{25} \left( \frac{NMHC^i}{NMHC} + \frac{OH^i}{OH} \right) \tag{12}$$

HO$_2$ is produced by reactions (1) H + O$_2$, (6) OH + O$_3$, (9) H$_2$O$_2$ + OH, (13) ClO + OH, (18) HNO$_4$, (24) NMHC + NO$_y$, (25) NMHC + OH and (26) NMHC + $hv$. However, H is not explicitly tagged in reaction (1). To be able to determine the HO$_2$ production by reaction (1) $R_1 \frac{H^i}{H}$, we apply the introduced H tagging (see Sect.3.4) and replace $\frac{H^i}{H}$ with equation (10).

Consequently, the HO$_2$ production per source category $i$ $ProdHO2^i$ is determined:

$$ProdHO_2^i = \frac{1}{2} R_6 \left( \frac{OH^i}{OH} + \frac{O_3^i}{O_3} \right) + \frac{1}{2} R_7 \left( \frac{OH^i}{OH} + \frac{O_3^i}{O_3} \right) + \frac{1}{2} R_9 \left( \frac{HO_2^i}{HO_2} + \frac{OH^i}{OH} \right) + R_{10} \frac{OH^i}{OH}$$
$$+ \frac{1}{2} R_{11} \left( \frac{CO^i}{CO} + \frac{OH^i}{OH} \right) + 0.94 \cdot R_{13} \frac{OH^i}{OH} + R_{18} \frac{NO_y^i}{NO_y} + \frac{1}{2} R_{24} \left( \frac{NMHC^i}{NMHC} + \frac{NO_y^i}{NO_y} \right)$$
$$+ \frac{1}{2} R_{25} \left( \frac{NMHC^i}{NMHC} + \frac{OH^i}{OH} \right) + R_{26} \frac{NMHC^i}{NMHC} + R_{31} \frac{NMHC^i}{NMHC} \tag{13}$$

The HO$_2$ loss is determined by reactions (3) HO$_2$ + HO$_2$, (4) HO$_2$ + O$_3$, (5) HO$_2$ + O($^3$P), (8) HO$_2$ + OH, (14) NO + HO$_2$, (15) NO$_2$ + HO$_2$, (23) NMHC + HO$_2$, (27) ClO + HO$_2$ and (29) BrO + HO$_2$. As HO$_2$ reacts with itself in reaction (3), $\frac{HO_2^i}{HO_2}$ is





counted twice. Hence, the HO$_2$ loss per source category $i$ $LossHO2^i$ is:

$$
\begin{aligned}
LossHO_2^i =& R_3\frac{HO_2^i}{HO_2} + \frac{1}{2}R_4\left(\frac{HO_2^i}{HO_2} + \frac{O_3^i}{O_3}\right) + \frac{1}{2}R_5\left(\frac{HO_2^i}{HO_2} + \frac{O_3^i}{O_3}\right) + \frac{1}{2}R_8\left(\frac{HO_2^i}{HO_2} + \frac{OH^i}{OH}\right) \\
&+ \frac{1}{2}R_{14}\left(\frac{NO_y^i}{NO_y} + \frac{HO_2^i}{HO_2}\right) + \frac{1}{2}R_{15}\left(\frac{NO_y^i}{NO_y} + \frac{HO_2^i}{HO_2}\right) + \frac{1}{2}R_{23}\left(\frac{NMHC^i}{NMHC} + \frac{HO_2^i}{HO_2}\right) \\
&+ R_{27}\frac{HO_2^i}{HO_2} + R_{29}\frac{HO_2^i}{HO_2}
\end{aligned}
\tag{14}
$$

Sect. 3.5 shows that the steady-state assumption for OH and HO$_2$ is justified when the rest terms $resOH$, $resHO_2$ and $resH$ are regarded. Therefore, the rest terms are divided by the number of source categories $n$ to add them to the contributions of a specific category $i$. In a steady-state, production of OH$^i$ and HO$_2^i$ equals the loss:

$$
ProdOH^i - LossOH^i + resOH/n = 0
\tag{15}
$$

$$
ProdHO_2^i - LossHO_2^i + resHO_2/n + resH/n = 0
\tag{16}
$$

The equations (15) and (16) can be rewritten as follows:

$$
0 = A^i - L^{OH}\frac{OH^i}{OH} + P^{OH}\frac{HO_2^i}{HO_2} + \frac{resOH}{n}
\tag{17}
$$

$$
0 = B^i + P^{HO_2}\frac{OH^i}{OH} - P^{HO_2}\frac{HO_2^i}{HO_2} + \frac{resHO_2}{n} + \frac{resH}{n}
\tag{18}
$$

with the variables $P^{OH}$, $L^{OH}$, $P^{HO_2}$, $L^{HO_2}$, $A^i$ and $B^i$ (compare to Grewe et al. (2017) equations (25) to (28)):

$$
P^{OH} = \frac{1}{2}R_4 + \frac{1}{2}R_5 + \frac{1}{2}R_{14} - \frac{1}{2}R_8
\tag{19}
$$

$$
L^{OH} = \frac{1}{2}R_6 + \frac{1}{2}R_7 + \frac{1}{2}R_8 + R_9 + R_{10} + \frac{1}{2}R_{11} + R_{12} + R_{13} + \frac{1}{2}R_{16} + \frac{1}{2}R_{17} + \frac{1}{2}R_{22} + \frac{1}{2}R_{25}
\tag{20}
$$

$$
P^{HO_2} = \frac{1}{2}R_6 + \frac{1}{2}R_7 + R_9 + R_{10} + \frac{1}{2}R_{11} + 0.94\cdot R_{13} + \frac{1}{2}R_{25} - \frac{1}{2}R_8
\tag{21}
$$

$$
L^{HO_2} = 2\cdot R_3 + \frac{1}{2}R_4 + \frac{1}{2}R_5 + \frac{1}{2}R_8 + \frac{1}{2}R_{14} + \frac{1}{2}R_{15} + \frac{1}{2}R_{23} + R_{27} + R_{29}
\tag{22}
$$

$$
\begin{aligned}
A^i =& 2\cdot R_2\frac{O_3^i}{O_3} + \frac{1}{2}R_4\frac{O_3^i}{O_3} + \frac{1}{2}R_5\frac{O_3^i}{O_3} + \frac{1}{2}R_{14}\frac{NO_y^i}{NO_y} + R_{20}\frac{NO_y^i}{NO_y} + R_{21}\frac{NO_y^i}{NO_y} \\
&- \frac{1}{2}R_6\frac{O_3^i}{O_3} - \frac{1}{2}R_7\frac{O_3^i}{O_3} - \frac{1}{2}R_{11}\frac{CO^i}{CO} - \frac{1}{2}R_{16}\frac{NO_y^i}{NO_y} - \frac{1}{2}R_{17}\frac{NO_y^i}{NO_y} - \frac{1}{2}R_{22}\frac{NMHC^i}{NMHC} - \frac{1}{2}R_{25}\frac{NMHC^i}{NMHC} \\
&+ \frac{resOH}{n}
\end{aligned}
\tag{23}
$$





$$B^i = \frac{1}{2}R_6\frac{O_3^i}{O_3} + \frac{1}{2}R_7\frac{O_3^i}{O_3} + \frac{1}{2}R_{11}\frac{CO^i}{CO} + R_{18}\frac{NO_y^i}{NO_y} + \frac{1}{2}R_{24}\left(\frac{NMHC^i}{NMHC} + \frac{NO_y^i}{NO_y}\right) + \frac{1}{2}R_{25}\frac{NMHC^i}{NMHC}$$
$$+ R_{26}\frac{NMHC^i}{NMHC} + R_{31}\frac{NMHC^i}{NMHC} - \frac{1}{2}R_4\frac{O_3^i}{O_3} - \frac{1}{2}R_5\frac{O_3^i}{O_3} - \frac{1}{2}R_{14}\frac{NO_y^i}{NO_y} - \frac{1}{2}R_{15}\frac{NO_y^i}{NO_y} - \frac{1}{2}R_{23}\frac{NMHC^i}{NMHC}$$
$$+ \frac{resHO_2}{n} + \frac{resH}{n} \tag{24}$$

Solving equations (17) and (18), we finally obtain for the contributions of a source category $i$ to the OH and $HO_2$ concentration (compare to Grewe et al. (2017) equations (29) and (30)):

$$\frac{OH^i}{OH} = \frac{A^i L^{HO_2} + B^i P^{OH}}{L^{OH} L^{HO_2} - P^{OH} P^{HO_2}} \tag{25}$$

$$\frac{HO_2^i}{HO_2} = \frac{A^i P^{HO_2} + B^i L^{OH}}{L^{OH} L^{HO_2} - P^{OH} P^{HO_2}} \tag{26}$$

These equations are implemented in the TAGGING submodel and EMAC and MECO(n) simulations after Sect. 2 are performed. The results for the OH and $HO_2$ contributions are analysed and compared with V1.0 in the following Section.

## 4 Results of model simulations

### 4.1 Contribution of short-lived species ($HO_x$)

Figure 2 and 3 show the zonal mean of the OH and $HO_2$ contributions for the ten source categories up to 200 hPa. The zonal mean of OH and $HO_2$ contributions from 1 to 200 hPa are shown in the appendix A (Fig. A1, A2). For the categories which are determined by anthropogenic emissions, such as "shipping", "road traffic" and "anthropogenic non-traffic", the maximum values of OH and $HO_2$ contributions occur in the lower troposphere in the Northern Hemisphere. This clearly shows that for the anthropogenic dominated categories the OH and $HO_2$ contributions are caused by the anthropogenic emissions. For the category "aviation", maximum OH contribution are found in the Northern Hemisphere between 200 and 250 hPa. However, $HO_2$ contribution has a minimum in this region and a maximum in the lower troposphere. The OH values for the categories "$CH_4$", "$N_2O$, "lightning" and "biogenic emissions" are largest in the upper troposphere. Most OH contributions of "biomass burning" are found in lower tropical troposphere. In contrast, negative values occurs in the upper tropical troposphere. Concerning the $HO_2$ contribution, the residual categories show a maximum in the tropical lower troposphere. In addition, the category "lightning" shows a strong $HO_2$ loss in the upper tropical troposphere which is caused by reaction (14).

The results obtained in this study are compared to the OH and $HO_2$ zonal profiles of V1.0 only in the troposphere up to 200 hPa. The $HO_x$ tagging mechanism V1.0 was only developed for the troposphere. Hence, a comparison in the stratosphere would not be reasonable. The relative differences of V1.0 towards V1.1 for the year 2010 are shown in Fig. 4 and 5. In general, contributions to OH concentrations of V1.1 are larger in the free troposphere and smaller in the boundary layer compared to V1.0. Also, the $HO_2$ contributions show a large increase over the whole troposphere. This overall shifts towards larger values



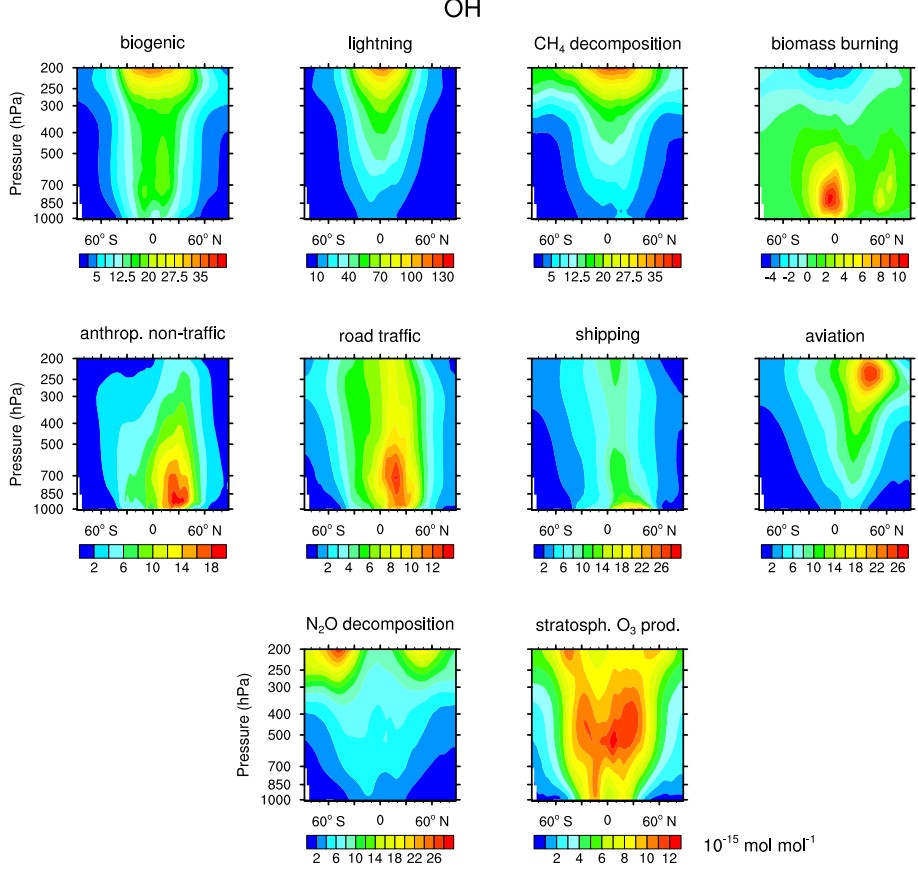

**Figure 2.** Contribution of ten source categories to OH in $10^{-15}$ mol mol$^{-1}$. Zonal mean of the year 2010 are shown. Simluation is performed with EMAC.

can be explained by the introduction of the rest terms: For each of the ten source category a tenth of the rest term is added to the OH and HO$_2$ contributions, which leads to larger values compared to V1.0.

V1.0 overestimated the contributions to OH in the boundary layer because only a few OH loss reactions were considered. The addition of new OH loss reactions such as reaction (10) cause a lower contributions to OH in V1.1. Moreover, V1.0 indirectly calculated the rate of reaction (22) by the production of CO. In comparison, V1.1 directly calculates the rate by including all reactions with NMHC (see Sect. 3.3). This also affects the change of OH contribution in the boundary layer.

For OH, the categories "lightning", "shipping" and "'aviation" show no large changes in the general pattern of the zonal means. The categories "anthropogenic non-traffic", "road traffic", "N$_2$O decomposition" and "stratospheric O$_3$ production" showed an OH loss in the upper tropical troposphere in V1.0 which is not visible in V1.1 anymore.

The "biomass burning" category shows large changes in pattern. The OH contributions strongly decreases in the upper and increases in the lower troposphere. Many new reactions of NMHC are added in V1.1. These reactions as well as the better representation of reaction rate (22) are responsible for the decrease of the minimum in the upper troposphere.





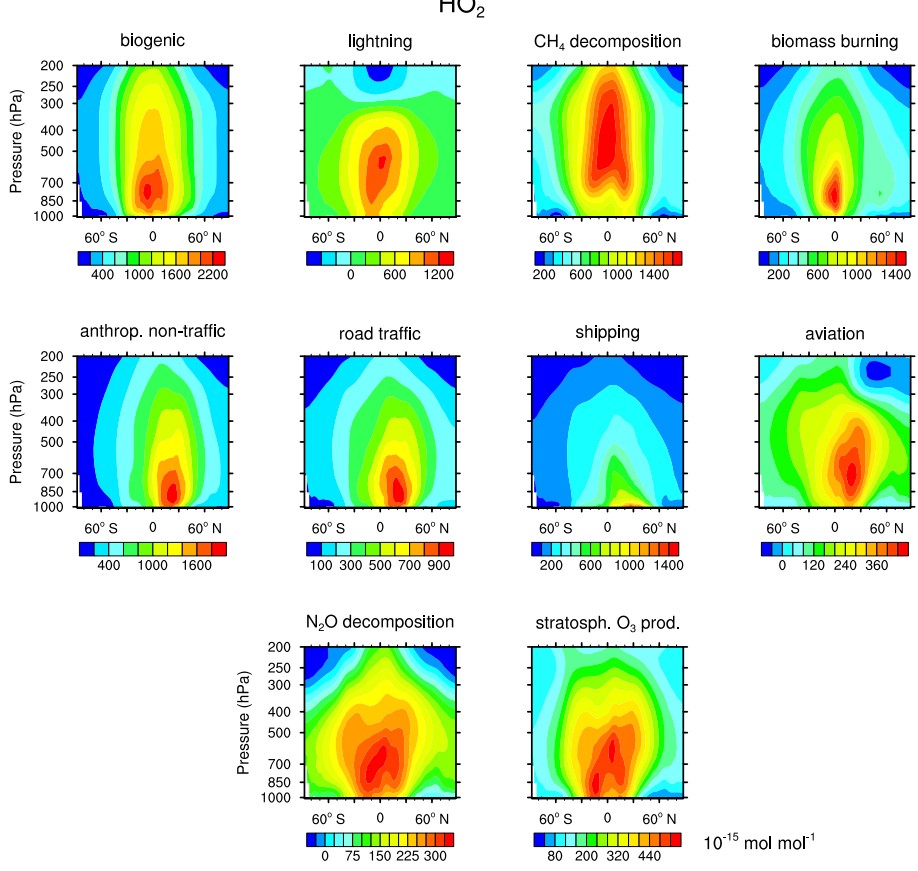

**Figure 3.** Contribution of ten source categories to $HO_2$ in $10^{-15}$ mol mol$^{-1}$. Zonal mean of the year 2010 are shown. Simluation is performed with EMAC.

Major changes are found in the categories "$CH_4$ decomposition" and "biogenic emissions". The OH contribution of both categories in V1.0 was negative over the whole atmosphere with a strong OH loss in the upper tropical troposphere. In V1.1, the patterns switch their signs: the contributions are positive with a maximum in the upper tropical troposphere. In both categories, NMHC are dominating. The reactions of NMHC with OH, $HO_2$ and $NO_y$ (reaction 22, 23 and 24) are important throughout the whole troposphere. V1.0 did not consider all reactions of NMHC with OH, $HO_2$ and $NO_y$ (see Sect. 3.3). Now, V1.1 regards all reactions of OH, $HO_2$ and $NO_y$ with NMHC. This causes large changes in pattern.

Considering the $HO_2$ contributions, no large changes in the general zonal pattern are found in the categories "biomass burning", "anthropogenic non-traffic", "road traffic" and "shipping". Large changes in pattern occur for the category "biogenic emissions" and "$CH_4$ decomposition". Especially in the lower troposphere, V1.0 showed a rather ragged structure while V1.1 presents a smooth pattern. No negative values occur anymore. Many new reactions are now considered in the $HO_x$ tagging mechanism and thus contribute to pattern changes. In particular for "$CH_4$ decomposition", the consideration of further reactions of NMHC producing $HO_2$ (reaction 25 and 26) largely contributes to the pattern change in $HO_2$.





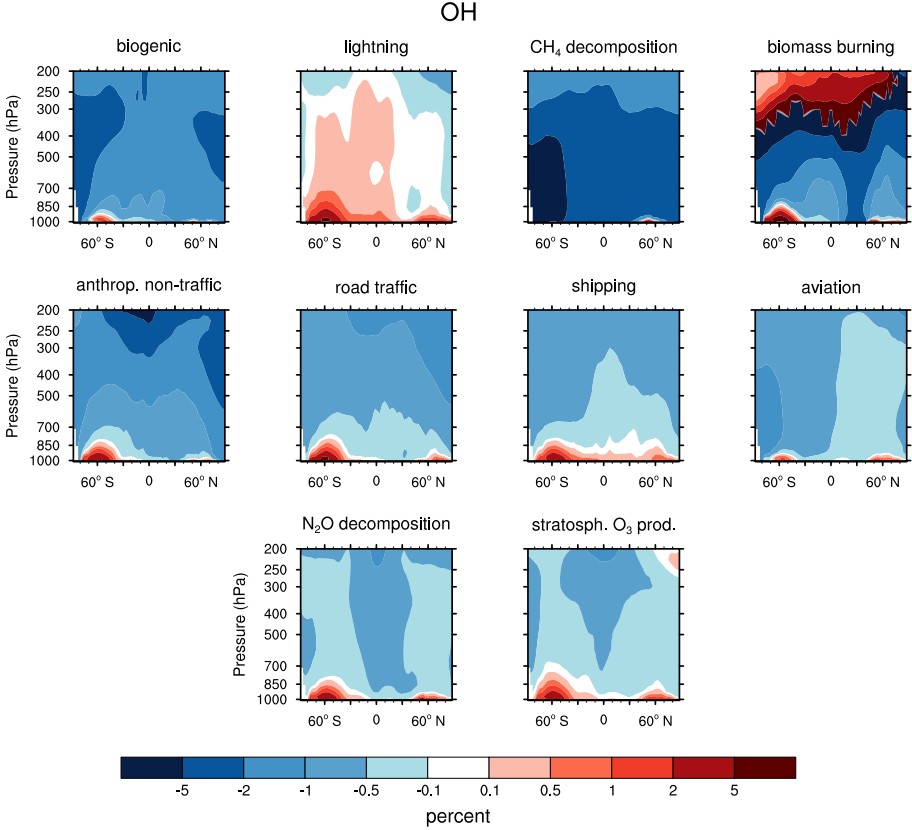

**Figure 4.** Relative differences of the OH contribution obtained by V1.0 towards the further developed $HO_x$ tagging mechanism V1.1. To be consisted, the same year 2010 is taken into consideration.

The larger values of "$N_2O$ decomposition" and "lightning" in the upper troposphere can be explained by several effects. Amongst others, the introduced H tagging contributes to a larger $HO_2$ production. Furthermore, mostly $NO_y$ emissions contribute to the categories "$N_2O$ decomposition" and "lightning". V1.1 regards many new reactions concerning $NO_y$. In particular, the photolysis of HONO and $HNO_3$ increases regionally the OH production. Via equation (26) the addition of these reactions
5   also causes the $HO_2$ increase in the upper troposphere.

The category "aviation" shows roughly the same pattern compared to V1.0. However, the $HO_2$ destruction in the flight path is not as pronounced any more. The sector "aviation" is dominated by $NO_y$ emissions. Considering reaction (18) in V1.1 adds an extra $HO_2$ source to this region. Thus the minimum is not as pronounced in V1.1 compared to V1.0.

The same effect can be seen in categories "road traffic", "shipping" and "anthropogenic non-traffic". In the boundary layer
10   at 40-60° N, V1.0 had a $HO_2$ minimum. The consideration of reaction (18) adds an extra $HO_2$ source and thus dissolves the alleged $HO_2$ reduction of V1.0.



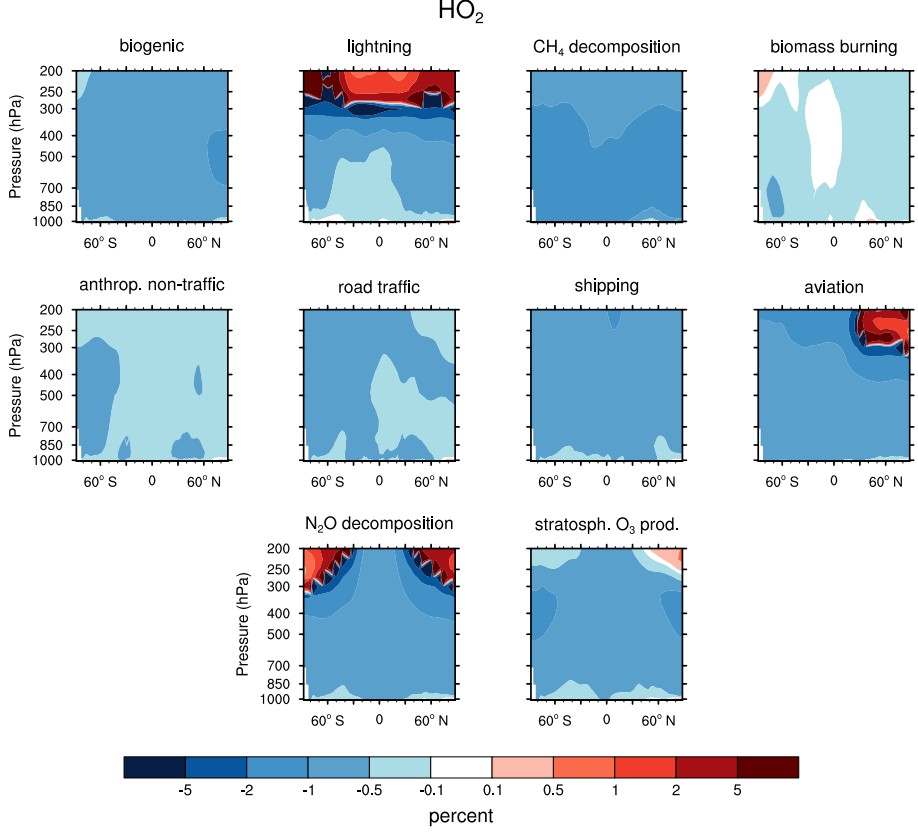

**Figure 5.** Relative differences of the HO$_2$ contribution obtained by V1.0 towards the further developed HO$_x$ tagging mechanism V1.1. To be consisted, the same year 2010 is taken into consideration.

To quantify the impact of the advanced HO$_x$ tagging mechanism on regional scale, Fig. 6 shows the contributions of ship emissions to OH and HO$_2$ in the boundary layer simulated with the high resolution model MECO(n) (see Sect. 2). The ship paths in the Atlantic, Mediterranean and Red Sea are clearly visible and lead to OH and HO$_2$ production along these paths. In the polluted area at the coast of Marseille the OH and HO$_2$ contributions are reduced, although NO$_y$ from shipping emission is larger than in the Mediterranean Sea. However, O$_3$ from shipping is also larger in this region. This can cause stronger HO$_2$ and OH loss via reaction (4) and (6) than in the Mediterranean Sea.

The tagging mechanism V1.0 (Grewe et al. (2017) their Fig. 6) showed negative HO$_2$ shipping contribution along the ship paths. This could be explained by reaction (14): NO destroys HO$_2$ and leads to negative contributions. However, in V1.1 HO$_2$ shipping contributions are positive. The contribution of ships to NO$_y$ is very large. Thus, the addition of reaction (15) and (18) causes the change of sign. Although, the reaction rate $R_{15}$ equals the rate $R_{18}$, more HO$_2$ is produced by ships than is destroyed: only half of the HO$_2$ which is destroyed by reaction (15) is added to the HO$_2$ destruction per category $LossHO_2^i$





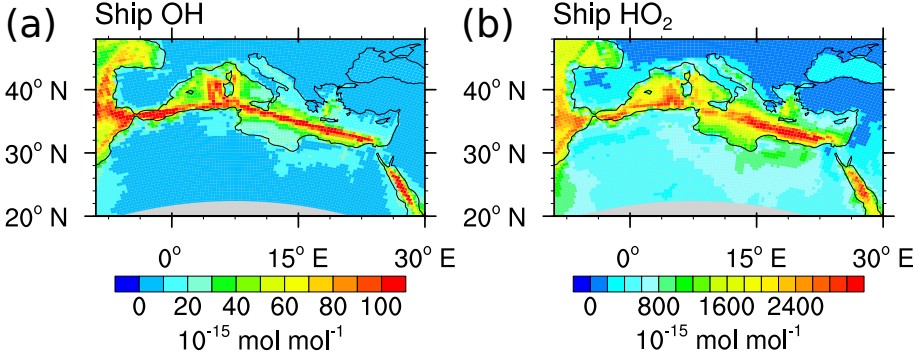

**Figure 6.** Contribution of shipping emissions to OH and $HO_2$ in $10^{-15}$ mol mol$^{-1}$. Monthly mean of ground level values in August 2007 are shown. Simluation is performed with MECO(n).

(eq. 14). In contrast, all $HO_2$ which is produced by reaction (18) is added to the $HO_2$ production per category $ProdHO_2^i$ (eq. 13). As more $HO_2$ of the shipping sector is produced than destroyed, a shift to positive values is caused.

To summarize, the contributions to OH and $HO_2$ concentrations show larger values in V1.1 compared to V1.0. This can be explained by the introduction of the rest terms. For OH, no changes are found in the categories "lightning", "shipping" and "aviation". However, large changes are found for "biomass burning", "$CH_4$ decomposition" and "biogenic emissions". For $HO_2$, no differences occur in the categories "biomass burning", "anthropogenic non-traffic", "road traffic" and "shipping". In comparison, the categories "biogenic emissions" and "$CH_4$ decomposition" differ strongly. The differences between the contributions of V1.1 and V1.0 can be traced back to the addition of certain reactions to the reduced reaction system considered in the $HO_x$ tagging mechanism.

## 4.2 Effects on long-lived species

The changes of $HO_x$ contributions feed backs to the long-lived tracer $O_3$, $NO_y$, CO, NMHC and PAN. Exemplary, Figure 7 shows the $O_3$ zonal mean for the ten source categories. Grewe et al. (2017) presents the same figure for the their $HO_x$ tagging mechanism (their Fig. 4). For consistency, we compare our results with the results of Grewe et al. (2017) only for the year 2010.

In general, no large differences between V1.1 and V1.0 for long-lived species are found. The category "biogenic emissions" and "$CH_4$ decomposition" show an increase in the tropical troposphere. "Stratospheric $O_3$ production" slightly increases in the Southern Hemisphere. Small $O_3$ changes are found for the categories "lighting", "biomass burning", "road traffic" and "$N_2O$ decomposition". Regarding the other long-lived species, CO from "biogenic emissions" and "$CH_4$ decomposition" decreases while CO from "lighting" increases in the Southern Hemisphere. The remaining sectors stay rather unchanged. Only the sectors "lighting", "$CH_4$ decomposition", "biomass burning" and "road traffic" vary little for PAN. $NO_y$ and NMHC show only minor changes. Even though, major differences in OH and $HO_2$ occur between V1.0 and V1.1, these do not have a large effect on the long-lived species.





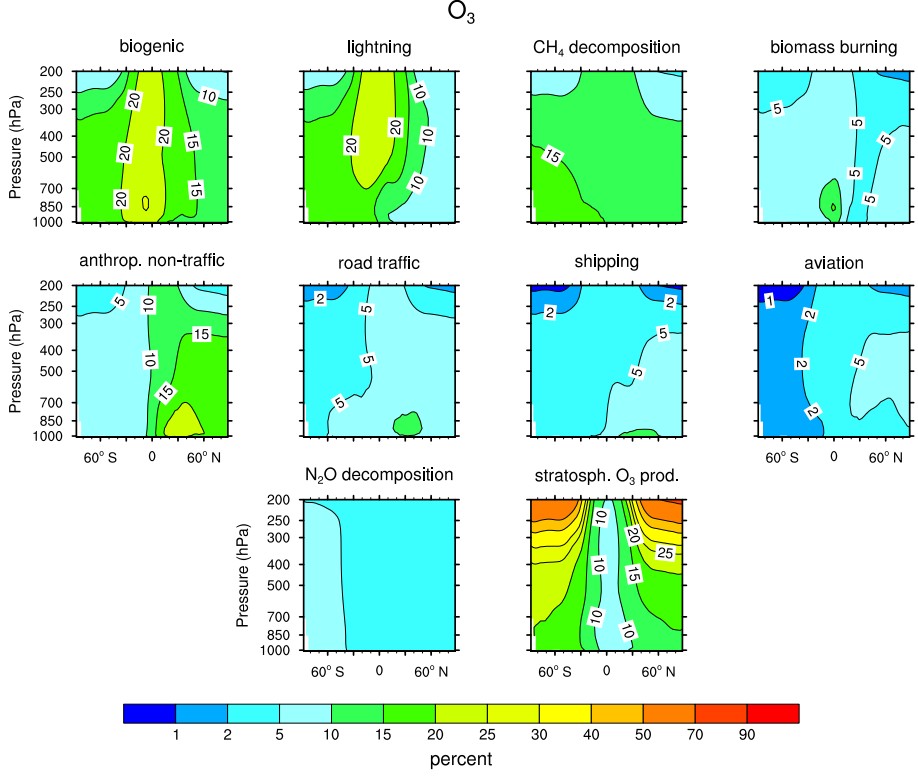

**Figure 7.** Annual mean contribution of ten source categories to $O_3$ concentration in %.

## 5 Conclusions

We present an extension of the $HO_x$ tagging mechanism described by Grewe et al. (2017). 15 new reactions producing and destroying $HO_x$ are added to tagging meachnism. In Grewe et al. (2017), the $HO_x$ tagging mechanism V1.0 was restricted to the troposphere only. We further include the reactions which are essential for $HO_x$ production and loss in the stratosphere. In

5   particular the production of $HO_2$ by H and $O_2$ and the reaction of OH and $HO_2$ with $O(^3P)$ are important in the stratosphere and are now taken into account. Moreover, we introduce an equivalent tagging mechanism to obtain the contributions to the H radical. This step is mandatory to fully account for the main $HO_2$ source: the reaction of H with $O_2$.

In V1.0, the budget of the sum of the $HO_x$ contributions and the total $HO_x$ concentration deviated by ca. 70 %. The addition of 15 new reactions to the reaction system leads to a better closure (deviation of 25 %). However, since we have to omit certain

10   reactions from the tagging mechanism, the production and loss rates are less balanced. Since this is a crucial assumption for the $HO_x$ tagging mechanism, we explicitly introduce rest terms to balance the deviation of $HO_x$ production and loss. The rest terms are equally distributed over the source categories. This leads to the closure of the budget. Thus, the tagging meachnism introduced by Grewe et al. (2010) operates not only for long-lived but also for short-lived species.



The advanced $HO_x$ tagging mechanism V1.1 was implemented in a global climate-chemistry model EMAC and in a regional model MECO(n). A 1-year simulation was performed in both model systems and compared to V1.0. The categories "lightning", "shipping" and "aviation" show no large changes in OH zonal pattern. Major changes in OH contributions are found in "biomass burning", "$CH_4$ decomposition" and "biogenic emissions". The $HO_2$ zonal pattern of the categories "biomass burning", "an-

thropogenic non-traffic", "road traffic" and "ship" do not differ from V1.0. However, the categories "$N_2O$ decomposition", "lightning" and "aviation" show some changes which could be traced back to the addition of individual reactions to the tagging mechanism. In general, an overall shift towards larger values is found in all categories which is caused by the rest terms. Little changes are found in $NO_y$, NMHC and PAN. However, $O_3$ presents certain changes in the tropical troposphere for the categories "biogenic emissions" and "$CH_4$ decomposition". Furthermore, CO shows some variations in the categories "biogenic

emissions", "$CH_4$ decomposition" and "lightning".

The further developed $HO_x$ tagging method can be used to identify the effect of anthropogenic emissions on the atmospheric composition. In particular, the contribution of emission sectors on the concentrations of OH and $HO_2$ in the troposphere and stratosphere can be achieved. This method will be applied for re-evaluating the impact of the traffic sector on the climate.

*Code availability.* The Modular Earth Submodel System (MESSy) is continuously further developed and applied by a consortium of insti-

tutions. The usage of MESSy and access to the source code is licensed to all affiliates of institutions, which are members of the MESSy Consortium. Institutions can become a member of the MESSy Consortium by signing the MESSy Memorandum of Understanding. More information can be found on the MESSy Consortium Web-site (http://www.messy-interface.org).

### Appendix A: $HO_x$ contributions in the stratosphere

Fig. A1 and A2 show the zonal mean of OH and $HO_2$ from 1 to 200 hPa. Note the logarithmic scale of the contour levels.

As OH concentration strongly raises with increasing height, so do the contributions to OH. The category "biomass burning" shows negative OH values in the tropopause region. In this region, also large CO values from "biomass burning" occur. CO effectively destroys OH by reaction (11) which causes this OH loss. The large negative minimum in the lower stratosphere of the category "stratospheric $O_3$ production" is mainly caused by the destruction of OH by $O_3$ (reaction 6).

The contributions of all categories to $HO_2$ in the stratosphere increases with height as well. The categories "biogenic emis-

sions", "lightning", "biomass burning", "anthropogenic non-traffic", "road traffic", "shipping" and "aviation" show a local maximum at around 5 hPa. Negative values occur in tropopause region for the category "lightning". This is induced by large values of $NO_y$ which mostly destroy $HO_2$ by reaction (14). The category "$N_2O$ decomposition" shows negative values in the lower stratosphere and a strong negative minimum at around 10 hPa which is also caused by reaction (14). The local maximum with positive $HO_2$ contributions indicates that in this region the $HO_2$ production via reaction (1) and (6) dominates the $HO_2$

loss via reaction (14).



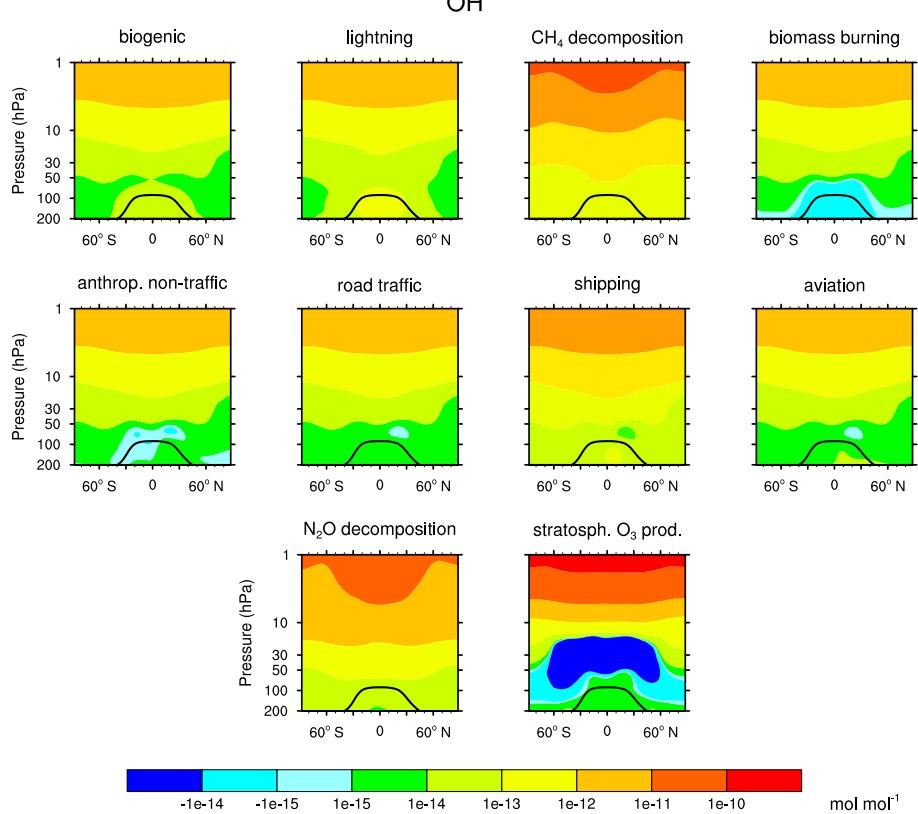

**Figure A1.** Contribution of ten source categories to OH in the stratosphere. Zonal mean of the year 2010 are shown. Black line indicates the tropopause. Simluation is performed with EMAC. Note the logarithmic scale of the contour levels.

*Competing interests.* There are no competing interests.

*Acknowledgements.* This study has been carried out in the framework of the project VEU2 funded by DLR. We used the NCAR Command Language (NCL) for data analysis and to create the figures of this study. NCL is developed by UCAR/NCAR/CISL/TDD and available online: doi: 10.5065/D6WD3XH5. We gratefully acknowledge the computer systems provided by the Deutsches Klimarechenzentrum (DKRZ) which we used for our simulations. We thank M. Righi from DLR for helpful comments.



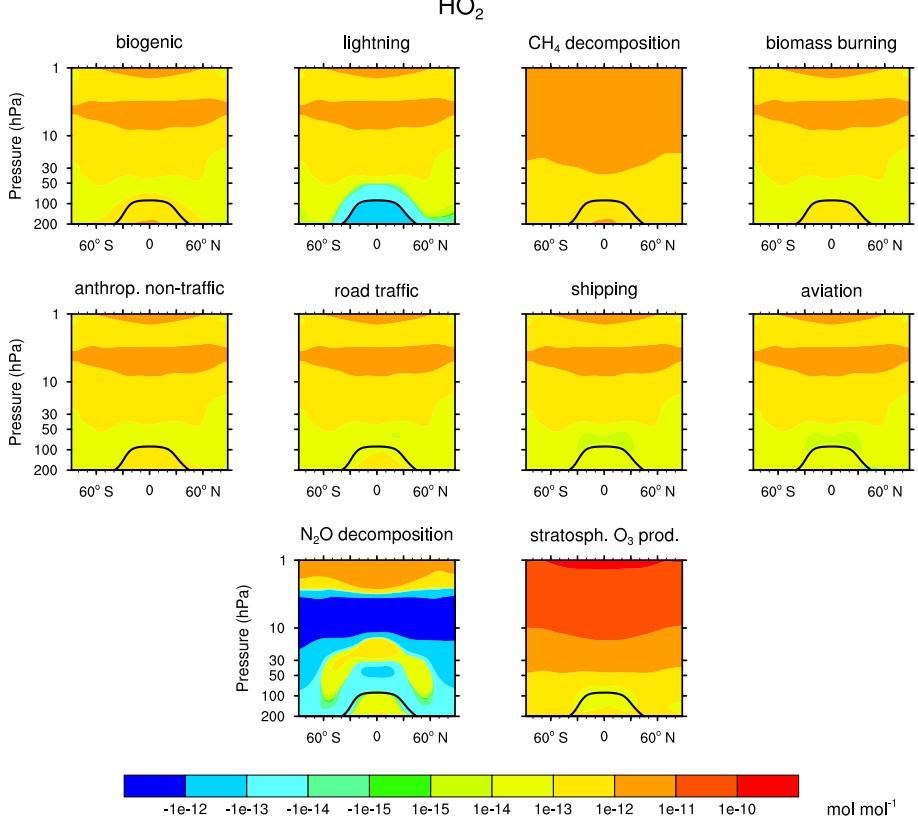

**Figure A2.** Contribution of ten source categories to HO$_2$ in the stratosphere. Zonal mean of the year 2010 are shown. Black line indicates the tropopause. Simluation is performed with EMAC. Note the logarithmic scale of the contour levels.

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
