# Peer review of "An advanced method of contributing emissions to short-lived chemical species (OH and $HO_2$ ): The TAGGING 1.1 submodel based on the Modular Earth Submodel System (MESSy 2.53)"

_Geoscientific Model Development, 2017_

## Short Comment (SC1) · 7 Nov 2017

Vanessa

The title suggests that TAGGING 1.1 is included in the MESSy 2.53 release and it is the version/configuration being used for the manuscript. Could you please explicitly state this in the Code Availability section? The title could also be read that TAGGING 1.1 is only an add-on, then access is not clear. Could you also please clarify how to access the code being used to obtain the results shown in the manuscript?

Many thanks.

Lutz Gross GMD Executive Editor

---

## Short Comment (SC2) · 8 Nov 2017

Dear Mr Gross,

You are right: the title can be read in two ways. The TAGGING 1.1 is developed based on MESSy 2.53. It will be available in the code starting from MESSy 2.54. We will modify the title and add an explanation in the Code Availability section in the revised version. Thank you very much for your advise.

Kind regards, Vanessa Rieger

---

## Referee Comment (RC1) · Anonymous Referee #1 · 22 Nov 2017

General comments

The present study by Rieger et al. presents an updated tagging algorithm for OH and HO2 which allows for an attribution of HOx to various emission categories. In contrast to the previous version V1.0 the new version takes into account tropospheric as well as stratospheric chemistry. Furthermore, the new scheme considers rest terms to take into account HOx production and loss reactions which are not explicitly tagged.

Since the tagging mechanism does not only consider primary contributions, but also

secondary effects via the long-lived species, the paper is rather difficult to read. I am sure for the authors, who are familiar with the tagging approach, it is clear what's going on, but for an unacquainted reader it is difficult to understand why an updated tagging method for HOx results in different contributions of, e.g., biomass burning emissions to ozone. Maybe a schematic would be helpful. Furthermore, the HOx tagging includes several assumptions and special cases. There are several open questions (details see below). My major concern is related to the steady-state assumption and the rest terms. Overall, I think the paper needs a clearer description of the method, a better justification of the assumptions and a more thoroughly explanation of the presented results.

In the introduction the authors argue that mitigation of climate change requires attribution of certain chemical trace gases to specific emission categories, and therefore propose the tagging approach. I wonder how robust the presented results are. For example, the presented model simulations do not consider direct CH4 emissions. I assume that tagging CH4 from the applied emission categories would have a large impact on the attribution of HOx to certain emissions. In my opinion it is inconsistent to consider CH4 decomposition as one category, while NOx, NMHCs etc. are split into road traffic, non-traffic etc. For an evaluation of the overall climate impact all traffic related emissions have to be tagged in the same manner. Therefore, I have doubts that the method is already applicable.

In the first sentence of Sect. 3.3 the authors state it is crucial that HOx production and loss of the reduced mechanism (almost) equal the complete HOx production and loss. Furthermore, steady state means to me that HOx production equals HOx loss. This is valid for the complete HOx chemistry derived from MECCA, but not or only partly for the reduced system. For example, stratospheric OH production of V1.1 deviates by 9% from the total stratospheric OH production and, maybe even more important, from the stratospheric OH loss of V1.1. Therefore, I do not agree with the conclusion at the end of section 3.3 that the steady-state assumption for the reduced reaction system V1.1 is justified. If the steady-state assumption was fulfilled, the rest terms to close

the budget would be needless. Furthermore, it is not clear from the paper that the steady-state assumption for H is valid, as claimed in Sect. 3.4. Table 3 presents the "main reactions of H" for the reduced system, but is this reaction system identical to the complete chemical mechanism in MECCA or only a subset?

Section 4 is mainly a description of the presented figures, but lacks explanations, for example for the differences between V1.0 and V1.1. The argumentation is often rather vague, namely that V1.1 considers now more reactions which contributes to the differences. That leaves the impression that the authors themselves do not fully understand the changed patterns. From what is written in the paper it is hard to understand the presented results and differences, but I think this is important to judge the performance and shortcomings of the tagging method. For example, what is the reason for the changes in the contribution of stratospheric O3 production to tropospheric OH? Furthermore, I am concerned about the HO2 shipping contribution discussed in Fig. 6. In this case the authors provide a clear explanation, but frankly speaking this example seems to show that the tagging method does not work. Two reactions (production and loss) with the same reaction rate, but only half of the loss is considered???

Specific comments

- P1, L11-14: As mentioned above, at first glance it is hard to understand why the tagging of HOx affects other tagged species. If you do not want to lose your readers right at the beginning, you should consider rewriting the last part of the abstract.

- P2, L8: For which specific environments? Please clarify.

- P4, L2/3: Why do you use a different time period for the global and the regional simulations (2007/2008 vs. 2009/2010)?

- P5, L7/8: I am not sure if I understand this statement correctly: In V1.0 the OH loss by the reaction with NMHCs was obtained from the total chemical CO production, no matter if OH was involved or not?

- P5, L17/18: Reaction rates depend on the concentrations of the reactants and via the rate coefficient often on temperature, and therefore vary in time and space. So how is the threshold reaction rate of 1e-15 mol/mol/s to interpret? Is that an annual and global mean value?

- P5, L21-31: What's the point of those two paragraphs? As long as you do not show any numbers, this part is rather vague.

- P5, L33: Why are reactions 19, 28 and 30 listed in Table 1, when they are not part of V1.1? That's inconsistent with the table caption.

- Table 1: How are NMHCs treated in the HOx tagging? Is there only one lumped NMHC or are the individual NMHCs treated separately? If it is a lumped NMHC, how often can it react with OH?

- Table 2: How exactly are the tropospheric and stratospheric production and loss rates calculated? I assume there is a kind of air mass weighting applied? mol/mol/s is a somehow weird unit, I would usually expect something like Tg(OH)/yr...

- P8, L2: It is true that the OH production for "all" differs only by 2% from the total production, but for "tag" the difference is about the same as for V1.0, namely around 11%. Which quantity is used for the tagging – "all" or "tag"?

- P9, L25: Why are the mentioned species not explicitly tagged? Please explain.

- P10, L8: I do not understand this sentence. Ratio of what to what?

- P10, L19: Why are H2O2 production and loss not balanced? Please explain. And HOx production and loss of the V1.1 reaction system are also not balanced, but nevertheless steady-state is assumed. This is inconsistent to me.

- P11, L14: In my view this is in contradiction to the statement at the end of Sect. 3.3 (-> steady-state assumption justified).

- P11, L28/29: Why are the rest terms equally distributed amongst the source cat-

egories (division by n) and not according to the contribution of the individual source categories to the total, e.g. OHi/OH? Couldn't it be that the linear distribution of the rest terms leads to an artificial exaggeration of a minor source category? I would be interested to see the contribution of the rest terms to the individual source categories.

- P14, Fig. 2: What is the reason for the different patterns in OH from anthrop. non-traffic, traffic and shipping? All three categories represent surface emissions.

- P16, L5/6: Please explain how the inclusion of more NMHC reactions leads to the changed pattern. In general, the treatment of NMHCs is not clear to me. Is there one lumped NMHC tracer?

- P17, L2/3: How do NOy emissions contribution to the category "N2O decomposition"?

- P19, L16/17: How does the HOx tagging affect stratospheric O3 production? Is it that HOx produced from ozone formed in the stratosphere leads to stratospheric ozone formation/destruction via the catalytic HOx cycles? But how would that fit with the family concept?

- Conclusions: The first paragraph has a lot of redundancy and should be shortened. This holds for several parts of the manuscript. And the second paragraph is more a repetition of the abstract than real conclusions.

- P21, L24-26: I assume that the local maximum around 5 hPa is a secondary effect via ozone?

- There is no single reference to the supplement in the manuscript. So why are those additional figures shown at all in the supplement? Honestly, I would prefer to see Fig. 1 and 2 from the supplement in the main paper instead of Fig. 2 – 4. That would make the comparison of V1.0 and V1.1 easier.

Technical comments

- Avoid overusing the definite article, e.g. P1, L21: "HOx impacts global warming and

local air quality. . ." or P2, L4: ". . .human impact on climate and air quality. . .."

- Caption Table 1, 3: "In the column "tropos." ("stratos.") reactions which are . . .."

- P7, L2: Do you mean Sect. 3.3 instead of 3.4?

- Eqn. 5: There is a mistake. For a unimolecular reaction, there is no LossBi, rather a ProdBi. And the reaction rate is reduced to R=kA, right? Should be mentioned.

- Eqns. 8 and 9: Is it possible that ProdHi and LossHi are swapped?

- Eqn. 18: I assume the third term on the right side should read -LHO2 instead of -PHO2

- Eqns. 23 and 24: Why are the rest terms includes in Ai and Bi? Doesn't that lead to a double-counting of the rest terms in eqn. 17 and 18?

- P14, L26: "This overall shift. . ."

- P21, L22: "large negative minimum" is a rather sloppy expression, please rephrase.

- Fig. 1: The reddish and pinkish lines are hard to distinguish, at least when printed. And I suggest to change the y-axis to 1.e-14 or 1.e-13 mol/mol/s to be consistent with the numbers given in the text.

- Caption Fig. 1: "(a) and (b) show the rates for the troposphere, . . ."

- Caption Fig. 2, 3, 6, A1, A2: "Zonal means . . . are shown." Simluation -> Simulation

- Caption Fig. 4, 5: consisted -> consistent

---

## Referee Comment (RC2) · Anonymous Referee #2 · 1 Dec 2017

**1   General Comments**

This paper presents an improvement to an existing and already implemented scheme, describing in detail the "tagging" of OH, HO$_2$, and H, by source sector. This is certainly a useful exercise. However, I found this paper confusing, as well as containing numerous errors. In its current form I am unable to recommend its publication in GMD.

[Figure]

**2 Specific Comments**

I found the description of the tagging (Section 3.4) very confusing - the terms "explicit tagging" and "specific tagging" are used, and seem to mean different things. Longer-lived species are also tagged by source region, but the paper does not make clear the difference in this tagging and the explicit or specific tagging mentioned. It is clear that this process is complicated and requires careful consideration, but it is not explained in a way that I could easily understand. Perhaps some sort of graphical description would be helpful here?

When extending the tagging scheme to include more reactions (listed in Table 1 of the paper), reactions 19 ($H_2O_2+h\nu \rightarrow 2OH$), 28 ($HOCl+h\nu \rightarrow OH+Cl$), and 30 ($HOBr+h\nu \rightarrow OH+Br$) are highlighted as being unable to be considered in the tagging scheme. However, the authors then include these reactions in Table 2 ("reduced - V1.1 all") and also in the line plots in Figure 1. They seem to make-up around 9% of the OH production rate, so I can see why they should be mentioned, but I was frustrated that they were given prominence over the "reduced - V1.1 tag" scheme, which is what was actually implemented in the model. Indeed, in Table 2 the OH loss and $HO_2$ production and loss rates are given alongside the "all" row and not the "tag", which I personally do not think is correct. I would see the "tag" scheme presented as the baseline, and the "all" is an extension to this. There is discussion in Section 3.3 about how good the "all" scheme is, but given it can't be used, why discuss it at all in this context?

I was also confused about the rest terms introduced in Section 3.5. I appreciated that closing the budget is desirable, but I do not believe that the text in Section 3.5 justifies or explains their introduction sufficiently, and they seem very artificial. Can the authors please expand on this justification and the necessity for having these terms?

Significant work is required by the authors to refine and clarify the manuscript. I suggest much more proof reading and editing are necessary prior to any resubmission.

**3  Technical Corrections**

1. I personally did not like the authors stating the species chemical formula after the name, without using either parentheses or parenthetical commas, e.g.

   *The radicals hydroxyl OH and hydroperoxyl $HO_2$ are crucial for the atmospheric chemistry.*

   rather than

   *The radicals hydroxyl (OH) and hydroperoxyl ($HO_2$) are crucial for the atmospheric chemistry.*

   or

   *The radicals hydroxyl, OH, and hydroperoxyl, $HO_2$, are crucial for the atmospheric chemistry.*

   This first format is used throughout the document (including the abstract). I would advise the authors to correct this to one of the others.

2. Page 1, line 16: remove "the" before "atmospheric chemistry".

3. Page 3, line 5: remove "the" before $HO_x$.

4. Page 4, line 6: could the authors please explain what a "cataster" is?

5. Page 4, lines 20-21: I would suggest either "The mechanism in V1.0" or "The V1.0 mechanism".

6. Page 5, lines 6-7: I don't quite understand what the authors mean by "Each reaction occurring in a simulation was precisely added up" in the context of the paragraph. Could the authors please re-phrase this?

7. Page 5, line 16: I would not use the phrase "boil down". I would suggest using "reduce" instead.

8. There is discussion in Section 3.2 about the relative contributions of various reactions to the OH and $HO_2$ budgets. It might be helpful to also visualise this, perhaps using bar- or pie-charts, perhaps in the Supplementary Information?

9. Page 13 equations 17, 18, 23, and Page 14 equation 24: Why does the term $resOH/n$ appear in both equation 17 and 23, and the terms $resHO_2/n$ and $resH/n$ appear in both 18 and 24. Looking at equations 15 and 16, shouldn't these terms appear only once each?

10. I was slightly frustrated by the use of different scales in the various sub-plots in Figure 2 (and also 3). While I appreciate there are orders of magnitude differences between various sectors, it would be helpful to have these all plotted on the same scale (with different common scales between Figures 2 and 3). I think that it would be helpful, as these are contrasted with Figures A1 and A2, which *do* have a common scale for all the sub-plots of each figure.

11. In Figures 4 and 5, is the use of the 0.1 to 0.5 (and -0.5 to -0.1) band useful? The authors explicitly discount changes this small, and would changes on these levels even be significant?

12. In Figures 4 and 5, could the authors explain the jagged feature seen in the OH biomass burning, the $HO_2$ $N_2O$ decomposition, and to a certain extent, the $HO_2$ lightning plots?

13. Page 19, line 4: I believe the authors mean "no large changes", not "no changes", as this is the wording they use in two other places in the manuscript.

14. Page 19, line 11: "long-lived tracers".

15. Page 19, line 11: I would not use "Exemplary", and would instead use "For example".

16. Page 19, last paragraph: Is this referencing the plots in the Supplementary Information? If so, please say so.

17. In the Supplementary Information, I would suggest labelling the figures as S1, S2 etc., especially since these figures should be referenced in the main text in some way, and it would be confusing otherwise.
* * *

---

## Author Comment (AC1) · 22 Feb 2018

**Response to Reviewer #1**

*We would like to thank the reviewer for thoroughly examining the manuscript. We gratefully incorporated the reviewer's comments which certainly improved our manuscript. Please find below our replies in italics and indented.*

**General comments**

The present study by Rieger et al. presents an updated tagging algorithm for OH and HO2 which allows for an attribution of HOx to various emission categories. In contrast to the previous version V1.0 the new version takes into account tropospheric as well as stratospheric chemistry. Furthermore, the new scheme considers rest terms to take into account HOx production and loss reactions which are not explicitly tagged.

Since the tagging mechanism does not only consider primary contributions, but also secondary effects via the long-lived species, the paper is rather difficult to read. I am sure for the authors, who are familiar with the tagging approach, it is clear what's going on, but for an unacquainted reader it is difficult to understand why an updated tagging method for HOx results in different contributions of, e.g., biomass burning emissions to ozone. Maybe a schematic would be helpful. Furthermore, the HOx tagging includes several assumptions and special cases. There are several open questions (details see below). My major concern is related to the steady-state assumption and the rest terms. Overall, I think the paper needs a clearer description of the method, a better justification of the assumptions and a more thoroughly explanation of the presented results.

> *We added some further explanation in the introduction about the interaction between short-lived and long-lived species to clarify the relations. As suggested by the reviewers, we also inserted a sketch explaining the interaction between long-lived and short-lived species (Fig. 1). We also restructured the manuscript by merging Section "Steady-state assumption" and "Closure of the budget". We further set the case "tag" as baseline for the manuscript as was suggested by reviewer #2. Moreover, we better justified the introduction of the rest terms in Sect. 3.4. We also tried to present the results in a better and more understandable way. We hope that this improves readability of the manuscript. Details of the changes are described below.*

In the introduction the authors argue that mitigation of climate change requires attribution of certain chemical trace gases to specific emission categories, and therefore propose the tagging approach. I wonder how robust the presented results are. For example, the presented model simulations do not consider direct CH4 emissions. I assume that tagging CH4 from the applied emission categories would have a large impact on the attribution of HOx to certain emissions. In my opinion it is inconsistent to consider CH4 decomposition as one category, while NOx, NMHCs etc. are split into road traffic, non-traffic etc. For an evaluation of the overall climate impact all traffic related emissions have to be tagged in the same manner. Therefore, I have doubts that the method is already applicable.

> *This is a good point. Indeed, it would be desirable to also tag $CH_4$. However, for the current state-of-the-art, this is not possible. Estimates of $CH_4$ lifetime are still quite uncertain. In particular, $CH_4$ lifetime against OH is generally underestimated by chemistry climate models (Jöckel et al., 2016). But $CH_4$ is an important greenhouse gas. Thus, to hold $CH_4$ lifetime on a reasonable level, $CH_4$ emissions are generally prescribed in state-of-the-art chemistry climate models. Therefore, considering the current treatment of $CH_4$ in chemistry climate model, it is not reasonable to tag $CH_4$.*

*Considering the tagging of $CH_4$ would of course change these categories where $CH_4$ emissions play a role. However, for those categories where $CH_4$ emissions do not play a major role, no large changes are expected. For example, for road traffic and shipping emissions, direct $CH_4$ emissions are not important. Consequently, no large changes are expected if the tagging of $CH_4$ would be included. Hence, the current implementation of the $HO_x$ tagging method enables to determine the contribution of traffic emissions to $HO_x$.*

*Note that we do not expect large changes in contributions from methane decomposition since this relies on $CH_4$ concentrations which are reasonable due to boundary conditions. However, a part of this decomposition will be allocated to other sources.*

In the first sentence of Sect. 3.3 the authors state it is crucial that HOx production and loss of the reduced mechanism (almost) equal the complete HOx production and loss. Furthermore, steady state means to me that HOx production equals HOx loss. This is valid for the complete HOx chemistry derived from MECCA, but not or only partly for the reduced system. For example, stratospheric OH production of V1.1 deviates by 9% from the total stratospheric OH production and, maybe even more important, from the stratospheric OH loss of V1.1. Therefore, I do not agree with the conclusion at the end of section 3.3 that the steady-state assumption for the reduced reaction system V1.1 is justified. If the steady-state assumption was fulfilled, the rest terms to close the budget would be needless.

*We agree that end of Sect.3.3 could be confusing. In fact, it referred to the reduced $HO_x$ reaction system taking all 30 reactions into account. For this case, the steady-state is valid for tropospheric OH as well as for tropospheric and stratospheric $HO_2$. Indeed, for the stratospheric OH, the production and the loss deviates by 9% and thus are not in steady-state. We have rewritten the paragraph and emphasized that the steady-state assumption for the reduced $HO_x$ reaction system V1.1 is not completely fulfilled.*

***Changes in manuscript:***
*Summing up, the reduced $HO_x$ reaction system V1.1 represents well the total $HO_x$ production and loss in the troposphere and stratosphere. V1.1 reproduces the $HO_x$ chemistry better than V1.0. However, OH production in troposphere and stratosphere as well as H loss in the stratosphere of V1.1 deviates from the total rates derived by MECCA. Thus, the state-state for the reduced $HO_x$ and H reaction system (Tables 1 and 2) is not completely fulfilled.*

Furthermore, it is not clear from the paper that the steady-state assumption for H is valid, as claimed in Sect. 3.4.

*We agree with the reviewer that we missed to explain this point. To show that steady-state of H is valid, we have added this information to Table 3 and discussed the steady-state of H in Sect. 3.4.*

***Changes in manuscript:***
*The reduced H reaction system in V1.1 (Table 2) represents the total H production and loss in the troposphere very well. However in the stratosphere, H loss in V1.1 deviates by 17 % from the total H loss.*

Table 3 presents the "main reactions of H" for the reduced system, but is this reaction system identical to the complete chemical mechanism in MECCA or only a subset?

*Table 3 shows a subset of the complete chemical mechanism in MECCA. The compete MECCA mechanism contains 15 reactions concerning H. To clarify the misunderstanding we have added an explanation in the text.*

***Changes in manuscript:***
*Table 3 presents the main reactions for H which still constitute a subset of full H chemistry implemented in MECCA.*

Section 4 is mainly a description of the presented figures, but lacks explanations, for example for the differences between V1.0 and V1.1. The argumentation is often rather vague, namely that V1.1 considers now more reactions which contributes to the differences. That leaves the impression that the authors themselves do not fully understand the changed patterns. From what is written in the paper it is hard to understand the presented results and differences, but I think this is important to judge the performance and shortcomings of the tagging method.

*Thank you for this comment. As you suggested we have replaced Fig. 2-4 in the manuscript by Fig. 1 and 2 from the Supplement. We agree that this makes the comparison between V1.0 and V1.1 easier.*
*We have stated the given explanations more precisely and hope that this can clarify the presented results. For example, we have merged the examples which are based on the same explanations.*

***Changes in manuscript:***
*The contribution of the category "aviation" to $HO_2$ in V1.1 shows roughly the same pattern compared to V1.0. However, the $HO_2$ destruction along the flight path is not as pronounced anymore which is caused by the inclusion of reaction (15) and (18) to V1.1. Reaction (15) adds the term $\frac{1}{2} R_{15} NO_y^j/NO_y$ to the $HO_2$ loss (eq. 12) and reaction (18) adds the term $R_{18} NO_y^j/NO_y$ to the $HO_2$ production (eq. 11). As reaction rate $R_{15}$ equals the rate $R_{18}$, this leads to a larger $HO_2$ production than $HO_2$ loss ($R_{18} NO_y^j/NO_y > \frac{1}{2} R_{15} NO_y^j/NO_y$). Consequently, the addition of reaction (15) and (18) to the reduced $HO_x$ reaction system V1.1 constitutes an extra $HO_2$ source.*
*Larger values of the categories "$N_2O$ decomposition" and "lightning" to $HO_2$ in the upper troposphere are explained by a larger $HO_2$ production in V1.1 compared to V1.0. The H tagging in V1.1 considers all relevant $HO_2$ sources (reaction (7), (10), (11) and (28)) leading to a larger $HO_2$ production. Also the addition of reactions (15) and (18) (explanation see above) as well as the addition of reaction (23) which considers more reactions than in V1.0 increase the $HO_2$ contribution of the categories "$N_2O$ decomposition" and "lightning".*
*Large changes in pattern are observed for the contributions of "biogenic emissions" and "$CH_4$ decomposition" to OH and $HO_2$ as well as for the contributions of "biomass burning" and "anthropogenic non-traffic" to OH. In V1.1, these categories mainly constitute a source of OH and $HO_2$ in the troposphere. The addition of reaction (24) and (25) to the reduced $HO_x$ reaction system V1.1 presents a $HO_2$ source increasing OH and $HO_2$ contributions. Furthermore, reactions of NMHC with OH, $HO_2$ and $NO_y$ (reaction 21, 22 and 23) are important throughout the whole troposphere. In contrast to V1.0, V1.1 considers all reactions of NMHC with OH, $HO_2$ and $NO_y$ (see Sect. 3.2) significantly changing the pattern of "biogenic emissions", "$CH_4$ decomposition", "biomass burning" and "anthropogenic non-traffic".*

For example, what is the reason for the changes in the contribution of stratospheric O3 production to tropospheric OH?

*Although the reactions of OH and $HO_2$ with $O(^3P)$ play only a minor role in the troposphere, their addition to reduced reaction system modifies the contributions to OH. Additionally, the re-establishment of the steady-state also increases the*

*contribution of stratospheric $O_3$ production to OH and thus causes the change in pattern.*

Furthermore, I am concerned about the HO2 shipping contribution discussed in Fig. 6. In this case the authors provide a clear explanation, but frankly speaking this example seems to show that the tagging method does not work. Two reactions (production and loss) with the same reaction rate, but only half of the loss is considered???

*The addition of new reactions to the reduced $HO_x$ reaction system changes the contribution of these sectors where the new reactions are relevant. For the category shipping, all reactions concerning $NO_y$ are relevant. Thus adding reaction (15) and (18) does change the contribution of shipping emissions to $HO_2$ significantly. We added further explanations in the text to clarify the change of sign due to addition of reactions (15) and (18).*
*This example shows that in V1.0 not all relevant reactions concerning $NO_y$ have been considered which leads to errors in the contribution calculations. In comparison, in V1.1, the tagging method overcomes these shortcomings.*

***Changes in manuscript:***
***Explanation:***
*The contribution of the category "aviation" to $HO_2$ in V1.1 shows roughly the same pattern compared to V1.0. However, the $HO_2$ destruction along the flight path is not as pronounced anymore which is caused by the inclusion of reaction (15) and (18) to V1.1. Reaction (15) adds the term ½ $R_{15}$ $NO_y^i/NO_y$ to the $HO_2$ loss (eq. 12) and reaction (18) adds the term $R_{18}$ $NO_y^i/NO_y$ to the $HO_2$ production (eq. 11). As reaction rate $R_{15}$ equals the rate $R_{18}$, this leads to a larger $HO_2$ production than $HO_2$ loss ($R_{18}$ $NO_y^i/NO_y$ > ½ $R_{15}$ $NO_y^i/NO_y$). Consequently, the addition of reaction (15) and (18) to the reduced $HO_x$ reaction system V1.1 constitutes an extra $HO_2$ source.*

***Example $HO_2$ shipping:***
*The change of sign is caused by the addition of reaction (15) and (18) to the reduced $HO_x$ reaction system V1.1 which constitutes a net $HO_2$ production leading to positive $HO_2$ contributions (explanation see above). The comparison shows that $HO_2$ contributions in V1.0 were systematically and erroneously underestimated.*

**Specific comments**

- P1, L11-14: As mentioned above, at first glance it is hard to understand why the tagging of HOx affects other tagged species. If you do not want to lose your readers right at the beginning, you should consider rewriting the last part of the abstract.

*Very good point. We definitely want to avoid losing readers already at the abstract. We added further explanations in the introduction and abstract of how the tagging of short-lived and long-lived species interacts. A detailed description of the tagging of long-lived species is found in Grewe et al. (2017).*

***Changes in manuscript:***
***Abstract:***
*As $HO_x$ reacts with ozone ($O_3$), carbon monoxide (CO), reactive nitrogen compounds ($NO_y$), non-methane hydrocarbons (NMHC) and peroxyacyl nitrates (PAN), the contributions to these species are also modified by the advanced $HO_x$ tagging method V1.1.*

***Introduction:***

*The contributions to long-lived and short-lived species are closely linked. For example, the reaction*
*OH +$O_3$ → $HO_2$ +$O_2$*
*involves the long-lived species $O_3$ and the short-lived species OH and $HO_2$. Hence, this reaction is considered in the implementation of the tagging method for long-lived and short-lived species. The contribution of, for example, shipping emissions to $O_3$ influences the contribution of shipping emissions to $HO_2$: the higher the contribution to $O_3$ is the more $HO_2$ is attributed to shipping emissions. Furthermore, OH from shipping emissions destroys $O_3$ and thus reduces the contribution of shipping emissions to $O_3$.*

- P2, L8: For which specific environments? Please clarify.

*Thank you for this hint, we added an example.*

***Changes in manuscript:***
*For certain environments, such as marine boundary layer, model studies compare well with measurements.*

- P4, L2/3: Why do you use a different time period for the global and the regional simulations (2007/2008 vs. 2009/2010)?

*We agree that this might we confusing. Therefore, we have repeated the EMAC simulation for the same time period 2007/2008 and adjusted the plots, tables and numbers in the paper to represent the year 2008. However, we leave the dates of Fig. 6 to be able to compare it with Grewe et al. (2017).*

- P5, L7/8: I am not sure if I understand this statement correctly: In V1.0 the OH loss by the reaction with NMHCs was obtained from the total chemical CO production, no matter if OH was involved or not?

*Yes, this is right. The reaction rate of OH + NMHC was determined only by the net CO production. We added some further explanation to clarify the calculation of the reaction rate in V1.0.*

***Changes in manuscript:***
*The reaction rate of OH with NMHC (reaction 21, Table 1) was determined via the production rates of CO by assuming that each reaction of OH with NMHC produces one CO molecule. This method neglects all intermediate oxidation reactions of NMHC and considers only these reactions when NMHC is finally oxidized to CO.*

- P5, L17/18: Reaction rates depend on the concentrations of the reactants and via the rate coefficient often on temperature, and therefore vary in time and space. So how is the threshold reaction rate of 1e-15 mol/mol/s to interpret? Is that an annual and global mean value?

*Thank you, we did not mention this in the text. The threshold is a tropospheric or stratospheric annual mean. We added an explanation in the text.*

***Changes in manuscript:***
*We consider only reactions with a tropospheric or stratospheric annual mean reaction rate larger than $10^{-15}$ mol $mol^{-1}s^{-1}$ (see Table 1).*

- P5, L21-31: What's the point of those two paragraphs? As long as you do not show any numbers, this part is rather vague.

*Thank you for pointing this out. We added the amounts of relative contributions of the mentioned reactions to the text.*

***Changes in manuscript:***
*The reactions which are important in the troposphere are indicated in Table 1. As stated above, reaction (1) of H and $O_2$ dominates the $HO_2$ production in the troposphere. It produces 49 % of tropospheric $HO_2$. In V1.0, only part of this $HO_2$ source was regarded (see Sect. 3.1). The most important $HO_2$ loss is the reaction with NO (reaction 14) followed by the reaction with itself producing $H_2O_2$ (reaction 3) which accounts for 32 % and 12 % of tropospheric $HO_2$ loss. The production via H2O and $O(^1D)$ produces about 21 % of tropospheric OH (reaction 2). The excited oxygen radical $(O(^1D))$ originates from the photolysis of $O_3$. Also reaction (14) of NO and $HO_2$ produces 32 % of tropospheric OH. OH is mostly destroyed by CO (reaction 11, 38 %) followed by NMHC (reaction 21, 25 %).*
*In the stratosphere different chemical reactions become important. Here, OH is mainly destroyed by $O_3$, producing 40 % of stratospheric $HO_2$. The reaction is partly counteracted by the reaction (14) which produces 21 % of OH and destroys 24 % of $HO_2$. Since large quantities of O3 are found in the stratosphere, $O_3$ or the excited oxygen radical $(O(^3P))$ destroys about 62 % of $HO_2$. Reactions with NMHC, CO and $CH_4$ play only a minor role in the stratosphere.*

- P5, L33: Why are reactions 19, 28 and 30 listed in Table 1, when they are not part of V1.1? That's inconsistent with the table caption.

*Yes indeed, there is contradicting. We have set the case "tag" as the baseline of the manuscript. Thus, Table 1 now represents only the reduced $HO_x$ reaction system V1.1 as the table caption says. The extra information is deleted and put into the appendix A.*

- Table 1: How are NMHCs treated in the HOx tagging? Is there only one lumped NMHC or are the individual NMHCs treated separately? If it is a lumped NMHC, how often can it react with OH?

*NMHC represents a chemical family which contains for example $CH_3OH$, $CH_3O_2$, $CH_3OOH$, HCHO, $C_2H_6$, $C_2H_4$ and $CH_3CO_3$. All species included in the chemical family NMHC are given in Table 1 in the Supplement of Grewe et al. (2017). The reaction rate of the reaction NMHC + OH $\rightarrow$ NMHC is determined by adding up all reaction rates of OH with the species of the family NMHC. We added further explanation to the text.*
*The number of reactions with OH is an interesting point, which we haven't followed yet. But that is something we might look into in future applications.*

***Changes in manuscript:***
*Reactions (21) to (25) involve the chemical family NMHC which contains several species such as formaldehyde (HCHO), ethylene $(C_2H_4)$ and propane $(C_3H_8)$. The rate for reaction (21) is determined by adding up the rates of all reactions of OH with each single species of the family NMHC.*

- Table 2: How exactly are the tropospheric and stratospheric production and loss rates calculated? I assume there is a kind of air mass weighting applied? mol/mol/s is a somehow weird unit, I would usually expect something like Tg(OH)/yr…

*The production and loss rates indicate annual means for the tropospheric and stratospheric domain. To calculate the means, each grid box is weighted with the corresponding air mass. We added this information to the table caption.*
*Concerning chemical reactions in the atmosphere, mol/mol/s is the usual unit for production and loss rates.*

***Changes in manuscript:***
*Annual mean of OH, $HO_2$ and H production and loss rates (air mass weighted)*

- P8, L2: It is true that the OH production for "all" differs only by 2% from the total production, but for "tag" the difference is about the same as for V1.0, namely around 11%. Which quantity is used for the tagging – "all" or "tag"?

*We agree that the discussion of the case "tag" and "all" was confusing. In the former version, the case "tag" is finally implemented in EMAC. For the current version, we set the implemented version "tag" as baseline for the paper. We hope this improves the readability of the paper.*

- P9, L25: Why are the mentioned species not explicitly tagged? Please explain.

*This is a good point. We missed to mention it. Due to limited computational resources, it is unfortunately not possible to tag all relevant species. This is also the reason why we also tag chemical families such as $NO_y$ and NMHC. We added an explanation in the text.*

***Changes in manuscript:***
*Due to limited computational resources, other species such as $H_2$, $H_2O_2$, $CH_4$, ClO and BrO are not tagged (as in V1.0).*

- P10, L8: I do not understand this sentence. Ratio of what to what?

*The sentence refers to the ratio $A^i/A$. To clarify this, we have rewritten the paragraph.*

***Changes in manuscript:***
*In reaction (1), neither H nor $O_2$ are tagged. To obtain the ratio $HO_2^i/HO_2$, we set up an extra tagging of H itself.*

- P10, L19: Why are H2O2 production and loss not balanced? Please explain. And HOx production and loss of the V1.1 reaction system are also not balanced, but nevertheless steady-state is assumed. This is inconsistent to me.

*$H_2O_2$ is not a radical, so we do not expect that production of $H_2O_2$ balances the loss of $H_2O_2$. In contrast, OH and $HO_2$ as well as H are all radicals which react very fast with many species in the atmosphere. Thus, steady state of OH, $HO_2$ and H is reached very fast. However, the reduced $HO_x$ reaction system V1.1 is indeed not balanced. Therefore, we introduce the rest terms.*

- P11, L14: In my view this is in contradiction to the statement at the end of Sect. 3.3 (-> steady-state assumption justified).

*We agree that the formulation at the end of Sect. 3.3 could be misleading. In the former version, end of Sect. 3.3 related to the case "all" which caused this misunderstanding. Since we now changed the manuscript and put the case "tag" as baseline of the manuscript, we modified the statements at the former end of Sect 3.3. So the introduction of the rest terms should be better justified now.*

- P11, L28/29: Why are the rest terms equally distributed amongst the source categories (division by n) and not according to the contribution of the individual source categories to the total, e.g. OHi/OH? Couldn't it be that the linear distribution of the rest terms leads to an artificial exaggeration of a minor source category? I would be interested to see the contribution of the rest terms to the individual source categories.

> *For example, a large part of the rest term for OH (resOH) originates from omitting the photolysis of $H_2O_2$ which produces OH. The question is now: From which source category does $H_2O_2$ come from? To which sector shall the produced OH attributed to? Since we don't know from which sector OH originates, we split it up equally among the sectors. Indeed, this method weights minor source categories stronger with the rest terms than major categories. However, a linear apportionment would assume that we know the origin of $H_2O_2$ what we indeed don't know.*
> *We provide figures showing resOH, resHO2 and resH in the supplement.*

- P14, Fig. 2: What is the reason for the different patterns in OH from anthrop. non-traffic, traffic and shipping? All three categories represent surface emissions.

> *Right, emissions of the sector anthropogenic non-traffic, road traffic and shipping are all surface emissions, but their composition and amounts are very different. For example, for the sector "anthropogenic non-traffic" CO emissions are dominating while for shipping, $NO_y$ emissions are dominating. Consequently, these three sectors cause a different response of the atmospheric chemistry and thus also different OH patterns (e.g. Hoor et al., 2009).*
> *Moreover, shipping emissions occur over the ocean where convection is less strong than over the land. This further explains that the contributions of shipping to OH are rather confined to the lower troposphere and do not reach as high into the free troposphere.*
>
> ***Changes to manuscript:***
> *The contributions vary among these categories of surface emissions as not only the amount but also the composition of the emissions differs.*

- P16, L5/6: Please explain how the inclusion of more NMHC reactions leads to the changed pattern. In general, the treatment of NMHCs is not clear to me. Is there one lumped NMHC tracer?

> *In the detailed chemistry scheme (MECCA) species are treated individually. For the tagging scheme they are lumped and the total reaction rate is taken as a sum from the detailed scheme. Hence in the tagging method, NMHC is a chemical family and includes species such HCHO, $C_2H_4$ and $C_3H_8$. The rates of reactions including NMHC are calculated by adding up all reaction rates from each single species in the family NMHC. Consequently, a change in the reaction rate also changes OH and $HO_2$ contributions.*
> *We added these explanations to the manuscript.*
>
> ***Changes to manuscript:***
> *Reactions (21) to (25) involve the chemical family NMHC which contains several species such as formaldehyde (HCHO), ethylene ($C_2H_4$) and propane ($C_3H_8$). The rate for reaction (21) is determined by adding up the rates of all reactions of OH with each single species of the family NMHC.*

- P17, L2/3: How do NOy emissions contribution to the category "N2O decomposition"?

*Thanks, this was unartfully expressed. Decomposition of $N_2O$ is a source of $NO_y$ in the stratosphere. We have thoroughly rewritten the paragraph.*

***Changes to manuscript:***
*Larger values of the categories "$N_2O$ decomposition" and "lightning" to $HO_2$ in the upper troposphere are explained by a larger $HO_2$ production in V1.1 compared to V1.0. The H tagging in V1.1 considers all relevant $HO_2$ sources (reaction (7), (10), (11) and (28)) leading to a larger $HO_2$ production. Also the addition of reactions (15) and (18) (explanation see above) as well as the addition of reaction (23) which considers more reactions than in V1.0 increase the $HO_2$ contribution of the categories "$N_2O$ decomposition" and "lightning".*

- P19, L16/17: How does the HOx tagging affect stratospheric O3 production? Is it that HOx produced from ozone formed in the stratosphere leads to stratospheric ozone formation/destruction via the catalytic HOx cycles? But how would that fit with the family concept?

*Ozone produced in the stratosphere does also influence the concentration of OH and HO2. $O_3$ reacts with OH and $HO_2$:*
*$OH + O_3 \rightarrow HO_2 + O_2$*
*$HO2 + O_3 \rightarrow OH + 2\ O_2$*
*As these reactions involve long-lived and short-lived species, they are regarded in the implementation of the tagging method for long-lived as well as in the tagging method for short-lived species.*
*Consequently, if the contribution of the sector "stratospheric $O_3$ production" to OH and $HO_2$ changes, this also affects the contribution to $O_3$.*

- Conclusions: The first paragraph has a lot of redundancy and should be shortened. This holds for several parts of the manuscript. And the second paragraph is more a repetition of the abstract than real conclusions.

*We have shortened the first part of the conclusion and added some further concluding thoughts.*
*We tried to avoid redundancy in the manuscript. However, we find it difficult to spot these redundancies because they also lead the reader through the manuscript. We tried to find a balance between redundancy and information necessary to understand the individual parts. We do not expect the reader to have read the whole manuscript, unlike the reviewer. Hence, a certain redundancy is required.*

***Changes to manuscript:***
*Please refer to the Sect. 5 Discussion and Conclusion.*

- P21, L24-26: I assume that the local maximum around 5 hPa is a secondary effect via ozone?

*Thank you for this comment. The photolysis of HOCl becomes important at around 5 hPa. Omitting this reaction from the tagging mechanism V1.1 leads to higher rest terms which are in turn responsible for the local maximum of $HO_2$ contributions at 5 hPa.*
*We added an explanation to the text.*

***Changes to manuscript:***
*The categories "biogenic emissions", "lightning", "biomass burning", "anthropogenic non-traffic", "road traffic", "shipping" and "aviation" show a local maximum at around 5 hPa which is caused by omitting the photolysis of HOCl (see Appendix A).*

- There is no single reference to the supplement in the manuscript. So why are those additional figures shown at all in the supplement? Honestly, I would prefer to see Fig. 1 and 2 from the supplement in the main paper instead of Fig. 2 – 4. That would make the comparison of V1.0 and V1.1 easier.

> *Thank you, this is a good point. We added references to the supplement. As you recommend, we also replaced Fig. 2-4 in manuscript by Fig. 1 and 2 from the supplement to enable a better comparison of V1.1 and V1.0.*

**Technical comments**

- Avoid overusing the definite article, e.g. P1, L21: "HOx impacts global warming and local air quality: : :" or P2, L4: ": : :human impact on climate and air quality: : :."

> *Thank you. Where possible, we tried to avoid the usage of "the".*

- Caption Table 1, 3: "In the column "tropos." ("stratos.") reactions which are : : :."

> *Changed.*

- P7, L2: Do you mean Sect. 3.3 instead of 3.4?

> *We actually wanted to refer to points 4. and 5. in Sect. 3.4. However, since we have restructured the manuscript, this is obsolete.*

- Eqn. 5: There is a mistake. For a unimolecular reaction, there is no LossBi, rather a ProdBi. And the reaction rate is reduced to R=kA, right? Should be mentioned.

> *Thank you for the correction. We added the adjusted reaction rate.*

- Eqns. 8 and 9: Is it possible that ProdHi and LossHi are swapped?

> *Thank you. We switched the labelling.*

- Eqn. 18: I assume the third term on the right side should read -LHO2 instead of -PHO2

> *Thank you for the correction. We changed it.*

- Eqns. 23 and 24: Why are the rest terms includes in Ai and Bi? Doesn't that lead to a double-counting of the rest terms in eqn. 17 and 18?

> *Thank you for this correction. This is a mistake. As the rest term are already mentioned in eqs. (17) and (18), they must not be repeated in eq. (23) and (24). We deleted them.*

- P14, L26: "This overall shift: : :"

> *Changed.*

- P21, L22: "large negative minimum" is a rather sloppy expression, please rephrase.

> *We modified the wording.*

***Changes to manuscript:***
*The large OH loss in the lower stratosphere*

- Fig. 1: The reddish and pinkish lines are hard to distinguish, at least when printed. And I suggest to change the y-axis to 1.e-14 or 1.e-13 mol/mol/s to be consistent with the numbers given in the text.

> *We deleted the figure 1 from the manuscript as the basic information is contained in Table 3 where we also included the production and loss rates of H tagging.*

- Caption Fig. 1: "(a) and (b) show the rates for the troposphere, : : :"

> *Changed.*

- Caption Fig. 2, 3, 6, A1, A2: "Zonal means : : : are shown." Simluation -> Simulation

> *Changed.*

- Caption Fig. 4, 5: consisted -> consistent

> *Changed.*

***References:***
*Grewe, V., Tsati, E., Mertens, M., Frömming, C., and Jöckel, P.: Contribution of emissions to concentrations: the TAGGING 1.0 submodel based on the Modular Earth Submodel System (MESSy 2.52), Geosci. Model Dev., 10, 2615-2633, https://doi.org/10.5194/gmd-10-2615-2017, 2017.*

*Hoor, P., Borken-Kleefeld, J., Caro, D., Dessens, O., Endresen, O., Gauss, M., Grewe, V., Hauglustaine, D., Isaksen, I. S. A., Jöckel, P., Lelieveld, J., Myhre, G., Meijer, E., Olivie, D., Prather, M., Schnadt Poberaj, C., Shine, K. P., Staehelin, J., Tang, Q., van Aardenne, J., van Velthoven, P., and Sausen, R.: The impact of traffic emissions on atmospheric ozone and OH: results from QUANTIFY, Atmos. Chem. Phys., 9, 3113-3136, https://doi.org/10.5194/acp-9-3113-2009, 2009.*

*Jöckel, P., Tost, H., Pozzer, A., Kunze, M., Kirner, O., Brenninkmeijer, C. A. M., Brinkop, S., Cai, D. S., Dyroff, C., Eckstein, J., Frank, F., Garny, H., Gottschaldt, K.-D., Graf, P., Grewe, V., Kerkweg, A., Kern, B., Matthes, S., Mertens, M., Meul, S., Neumaier, M., Nützel, M., Oberländer-Hayn, S., Ruhnke, R., Runde, T., Sander, R., Scharffe, D., and Zahn, A.: Earth System Chemistry integrated Modelling (ESCiMo) with the Modular Earth Submodel System (MESSy) version 2.51, Geosci. Model Dev., 9, 1153-1200, https://doi.org/10.5194/gmd-9-1153-2016, 2016.*

---

## Author Comment (AC2) · 22 Feb 2018

*We would like to thank the reviewer for the helpful comments on the manuscript. It helped a lot to improve our manuscript and to increase the readability. Please find below our replies in italics and indented.*

**1 General Comments**

This paper presents an improvement to an existing and already implemented scheme, describing in detail the "tagging" of OH, $HO_2$, and H, by source sector. This is certainly a useful exercise. However, I found this paper confusing, as well as containing numerous errors. In its current form I am unable to recommend its publication in GMD.

> *Based on the reviewer's comments, we thoroughly revised and restructured the manuscript. We agree with the reviewer that the case "tag" as baseline is better suited and we adapted the manuscript accordingly. In addition, we modified the presented figures and put the contour levels on a common scale, as suggested by the reviewer. We think that the method and results are now represented in a better way.*

**2 Specific Comments**

I found the description of the tagging (Section 3.4) very confusing - the terms "explicit tagging" and "specific tagging" are used, and seem to mean different things. Longer-lived species are also tagged by source region, but the paper does not make clear the difference in this tagging and the explicit or specific tagging mentioned. It is clear that this process is complicated and requires careful consideration, but it is not explained in a way that I could easily understand. Perhaps some sort of graphical description would be helpful here?

> *Obviously the text was misleading. There is no explicit or specific tagging. There is only one tagging method which we are using. This tagging method along with different assumptions based on the lifetime of the regarded species leads to different implementations. We changed the corresponding wordings to clarify this.*
> *We added further explanations on how the tagging of the short-lived and long-lived species influence each other in the introduction. As suggested by the reviewers, we also inserted a sketch explaining the interaction between long-lived and short-lived species (Fig. 1). The implementation of the long-lived tagging is explained in detail in Grewe et al. (2017).*

> **Changes in manuscript:**
> *The contributions to long-lived and short-lived species are closely linked. For example, the reaction*
> *$OH + O_3 \rightarrow HO_2 + O_2$*
> *involves the long-lived species $O_3$ and the short-lived species OH and $HO_2$. Hence, this reaction is considered in the implementation of the tagging method for long-lived and short-lived species. The contribution of, for example, shipping emissions to $O_3$ influences the contribution of shipping emissions to $HO_2$: the higher the contribution to $O_3$ is the more $HO_2$ is attributed to shipping emissions. Furthermore, OH from shipping emissions destroys $O_3$ and thus reduces the contribution of shipping emissions to $O_3$.*

When extending the tagging scheme to include more reactions (listed in Table 1 of the paper), reactions 19 ($H_2O_2 + hv \rightarrow 2OH$), 28 ($HOCl + hv \rightarrow OH + Cl$), and 30 ($HOBr + hv \rightarrow$

OH+Br) are highlighted as being unable to be considered in the tagging scheme. However, the authors then include these reactions in Table 2 ("reduced - V1.1 all") and also in the line plots in Figure 1. They seem to make-up around 9% of the OH production rate, so I can see why they should be mentioned, but I was frustrated that they were given prominence over the "reduced - V1.1 tag" scheme, which is what was actually implemented in the model. Indeed, in Table 2 the OH loss and HO2 production and loss rates are given alongside the "all" row and not the "tag", which I personally do not think is correct. I would see the "tag" scheme presented as the baseline, and the "all" is an extension to this. There is discussion in Section 3.3 about how good the "all" scheme is, but given it can't be used, why discuss it at all in this context?

> *Thank you for this recommendation. We changed the manuscript and set the case "tag", which is finally implemented in EMAC, as baseline of the manuscript. The moved the explanation about omitting certain reactions in the appendix. We hope the manuscript gained more readability.*

I was also confused about the rest terms introduced in Section 3.5. I appreciated that closing the budget is desirable, but I do not believe that the text in Section 3.5 justifies or explains their introduction sufficiently, and they seem very artificial. Can the authors please expand on this justification and the necessity for having these terms?

> *Thank you for this comment. We agree that the justification was not comprehensive. The steady-state assumption is the basic principle of the tagging method for short-lived species. As we consider a reduced $HO_x$ reaction system, the steady-state between production and loss is not fulfilled. To re-establish steady-state, we introduce the rest terms.*
> *We restructured the Sections "Steady-state assumption" and "Closure of the budget" in the manuscript and merged them together. We also added the above explanation to better justify the rest terms.*
>
> **Changes in manuscript:**
> *Thus, the state-state for the reduced $HO_x$ and H reaction system (Tables 1 and 2) is not completely fulfilled.*
> *But steady-state between production and loss is crucial for the tagging method for short-lived species. To re-establish steady-state, it would be necessary to include the complete $HO_x$ and H chemistry in the tagging method. However, this is not possible as the tagging method does not apply to all reactions of the $HO_x$ and H chemistry (for examples see Appendix A). Consequently, we introduce rest terms resOH, resHO2 and resH for OH, $HO_2$ and H to compensate for the deviations from steady-state.*

Significant work is required by the authors to refine and clarify the manuscript. I suggest much more proof reading and editing are necessary prior to any resubmission.

> *We thoroughly edited the manuscript based on the reviewer comments. We hope that it now better suits the reviewer's expectations.*

**3 Technical Corrections**

1. I personally did not like the authors stating the species chemical formula after the name, without using either parentheses or parenthetical commas, e.g.
    *The radicals hydroxyl OH and hydroperoxyl HO2 are crucial for the atmospheric chemistry.*
    rather than
    *The radicals hydroxyl (OH) and hydroperoxyl (HO2) are crucial for the atmospheric*

*chemistry.*
or
*The radicals hydroxyl, OH, and hydroperoxyl, HO$_2$, are crucial for the atmospheric chemistry.*
This first format is used throughout the document (including the abstract). I would advise the authors to correct this to one of the others.

*Thank you. We changed the notation to parentheses.*

2. Page 1, line 16: remove "the" before "atmospheric chemistry".

   *Done.*

3. Page 3, line 5: remove "the" before HO$_x$.

   *Done.*

4. Page 4, line 6: could the authors please explain what a "cataster" is?

   *We changed the word to inventory.*

5. Page 4, lines 20-21: I would suggest either "The mechanism in V1.0" or "The V1.0 mechanism".

   *We changed the wording.*

6. Page 5, lines 6-7: I don't quite understand what the authors mean by "Each reaction occurring in a simulation was precisely added up" in the context of the paragraph. Could the authors please re-phrase this?

   *We have reformulated the corresponding sentences.*

   ***Changes in manuscript:***
   *Most reaction rates used in the tagging method corresponds to the production and loss rates directly provided by the chemical scheme MECCA of EMAC.*

7. Page 5, line 16: I would not use the phrase "boil down". I would suggest using "reduce" instead.

   *We changed the word.*

8. There is discussion in Section 3.2 about the relative contributions of various reactions to the OH and HO$_2$ budgets. It might be helpful to also visualise this, perhaps using bar- or pie-charts, perhaps in the Supplementary Information?

   *Thank you for this hint. We added the amounts of the relative contributions of the mentioned reactions to the text.*

   ***Changes in manuscript:***
   *The reactions which are important in the troposphere are indicated in Table 1. As stated above, reaction (1) of H and O$_2$ dominates the HO$_2$ production in the troposphere. It produces 49 % of tropospheric HO$_2$. In V1.0, only part of this HO$_2$ source was regarded (see Sect. 3.1). The most important HO$_2$ loss is the reaction with NO (reaction 14) followed by the reaction with itself producing H$_2$O$_2$ (reaction 3) which accounts for 32 % and 12 % of tropospheric HO$_2$ loss. The production via H2O and O($^1$D) produces about*

*21 % of tropospheric OH (reaction 2). The excited oxygen radical ($O(^1D)$)) originates from the photolysis of $O_3$. Also reaction (14) of NO and $HO_2$ produces 32 % of tropospheric OH. OH is mostly destroyed by CO (reaction 11, 38 %) followed by NMHC (reaction 21, 25 %).*

*In the stratosphere different chemical reactions become important. Here, OH is mainly destroyed by $O_3$, producing 40 % of stratospheric $HO_2$. The reaction is partly counteracted by the reaction (14) which produces 21 % of OH and destroys 24 % of $HO_2$. Since large quantities of O3 are found in the stratosphere, $O_3$ or the excited oxygen radical ($O(^3P)$)) destroys about 62 % of $HO_2$. Reactions with NMHC, CO and $CH_4$ play only a minor role in the stratosphere.*

9. Page 13 equations 17, 18, 23, and Page 14 equation 24: Why does the term resOH=n appear in both equation 17 and 23, and the terms resHO2=n and resH=n appear in both 18 and 24. Looking at equations 15 and 16, shouldn't these terms appear only once each?

   *Yes, this is right. We deleted them in eqs. (23) and (24).*

10. I was slightly frustrated by the use of different scales in the various sub-plots in Figure 2 (and also 3). While I appreciate there are orders of magnitude differences between various sectors, it would be helpful to have these all plotted on the same scale (with different common scales between Figures 2 and 3). I think that it would be helpful, as these are contrasted with Figures A1 and A2, which do have a common scale for all the sub-plots of each figure.

    *Thank you for pointing this out. We changed these figures and put them on a common scale.*

11. In Figures 4 and 5, is the use of the 0.1 to 0.5 (and -0.5 to -0.1) band useful? The authors explicitly discount changes this small, and would changes on these levels even be significant?

    *This is a good point. We deleted these figures and replaced them to a direct comparison with V1.0 as it was recommended by reviewer #1.*

12. In Figures 4 and 5, could the authors explain the jagged feature seen in the OH biomass burning, the HO2 N2O decomposition, and to a certain extent, the HO2 lightning plots?

    *We exchanged these figures to a direct comparison to V1.0 as it was recommended by reviewer #1. The jagged features resulted from divisions with small numbers.*

13. Page 19, line 4: I believe the authors mean "no large changes", not "no changes", as this is the wording they use in two other places in the manuscript.

    *Yes, we mean "no large changes". So we changed it. Thank you.*

14. Page 19, line 11: "long-lived tracers".

    *Thank you for this hint. We corrected it.*

15. Page 19, line 11: I would not use "Exemplary", and would instead use "For example".

    *We changed it.*

16. Page 19, last paragraph: Is this referencing the plots in the Supplementary Information? If so, please say so.

    *Yes, indeed. We included the corresponding references.*

17. In the Supplementary Information, I would suggest labelling the figures as S1, S2 etc., especially since these figures should be referenced in the main text in some way, and it would be confusing otherwise.

    *This is a good point. We changed the labels of the supplement.*

---

## Author Comment (AC3) · 22 Feb 2018

Dear Mr Gross,

Thank you very much for your comment. I discussed the topic with my co-authors and we would like to leave the title as it is because it corresponds to the title of the paper "Contribution of emissions to concentrations: the TAGGING 1.0 submodel based on the Modular Earth Submodel System (MESSy 2.52)" by Volker Grewe, Eleni Tsati, Mariano Mertens, Christine Frömming, and Patrick Jöckel (GMD, 2017) on which our

manuscript is based on. However, we added further explanation to the manuscript to clarify that TAGGING 1.1 bases on MESSy 2.53 and will be included in MESSy 2.54 (Sect. 2) and in the Code Availability Section. Furthermore, we stated the code access in the Code Availability Section.

Kind regards, Vanessa Rieger
* * *

---

## Author Response (AR2)

**Institute of Atmospheric Physics/Earth-System-Modelling**

[Figure]

DLR e. V.   Institute of Atmospheric Physics/Earth-System-Modelling
            Oberpfaffenhofen, 82234 Weßling

Your reference

Your letter of

Our reference

Your correspondent   Vanessa Rieger

Telephone +49 8153 28-   1812
Telefax +49 8153 28-

E-mail   vanessa.rieger@dlr.de

19 April 2018

**Response to Editor**

Dear Mr Morgenstern,

We thoroughly revised the manuscript "An advanced method of contributing emissions to short-lived chemical species (OH and $HO_2$): The TAGGING 1.1 submodel based on the Modular Earth Submodel System (MESSy 2.53)" according to the reviews. We particularly considered the comments of reviewer #1 and hopefully we could resolve his concerns.
We hope to meet now the expectations of the reviewers and of GMD.

Yours sincerely,

Vanessa Rieger

Attachments:
- Response to Reviewer #1
- Response to Reviewer #2
- Marked-up manuscript

**Response to Reviewer #1**

**General comments**

The manuscript has significantly improved compared to the previous draft, in particular the readability. I still have some general doubts about the usefulness of the approach and the combined tagging of long- and short-lived species. However, given that the method and the caveats are now better explained, I think the paper is now publishable. I have some few, but minor suggestions for further clarification.

*Thank you very much for helping us to improve the manuscript. Please find the answers to your specific comments below.*

**Specific comments**

- p2, l12-16: I think a discussion about OH variability and methyl chloroform has to cite also the work by Stephen Montzka et al., e.g. Science (2011).

*Thank you for pointing out this work. We added it to the discussion about measuring OH.*

- p2, l22: Grewe et al. (2017)

*Thank you. We changed the text accordingly.*

- Fig.1: I think Figure 1 could be improved by adding the tagged emission categories. As far as I understand the combination of long- and short-lived species, the "starting point" of the tagging is ozone. That should be made visible in Fig. 1.

*As suggested by the reviewer, we have restructured Fig. 1 and put ozone in the center of the sketch. We also added the emissions contributing to $NO_y$, NMHC and CO to the figure. However, we refrain from adding all ten source categories to the figure as this would make the figure very confusing.*

- p5, l14-17: Ozone itself is not emitted. Please be more precise in your wording. Furthermore, it would be nice to have at least one sentence about which emitted species are distinguished. As I understand from the conclusions (p19, l11) the tagging considers NOy, CO and NMHC emissions. NOy in total, or NO and NO2 separately?

*This is a good point. We changed the text.*
*As suggested by the reviewer, we added an explanation about emitted species. Only NO and $NO_2$ are emitted which are accounted to $NO_y$. Emissions of CO contribute to CO concentration and emissions of e.g. $C_2H_4$, $C_3H_6$ or HCHO contribute to NMHC concentration.*

*Changes in manuscript:*
*For example, the concentration of $O_3$ is split up into $O_3$ produced by anthropogenic non-traffic (e.g. industry) emissions…*

*Emissions of NO and $NO_2$ contribute to $NO_y$ concentration, while emissions of e.g. $C_2H_4$, $C_3H_6$ or HCHO account to NMHC concentration.*

- p8, l26: … to the tagging method…

*We changed the text accordingly.*

- p8, l14: Why is reaction 14 listed as an example? Is NO not tagged?

*Thank you for this comment. This was a mistake. We replaced the example with reaction 13.*

- section 3.4: I really appreciate that the authors added this section about the steady-state assumption. However, I would prefer a slightly more neutral or scientific wording when it comes to the agreement between reduced and full chemical scheme. It is no big surprise that in v1.1 the sum of OH and HO2 balances the total concentrations after introducing the rest terms. That is the job of the rest terms.

*As recommended by the reviewer, we changed the discussion about the closure of the budget to a more neutral wording.*

*Changes in manuscript:*
*In V1.1, the sum of OH and HO2 now balances the total OH and HO2 concentrations. The deviations are negligible (below $10^{-3}$ %).*

I would be also interested to see how the rest terms affect the conclusions about the contribution of certain source categories, for example in the tropical lower troposphere, where the OH and HOx rest terms are largest.

*We added the zonal means of OH and $HO_2$ contributions for a simulation when the rest terms are neglected to the Supplement (Fig. S7 – S8). The comparison of these figures to OH and $HO_2$ contributions including the rest terms (Fig. 2 – 3 in Manuscript) shows that the general pattern is not affected by the rest terms. Only the OH contribution of the sectors "biomass burning" and "anthropogenic non-traffic" presents some deviations in the upper troposphere. In this region, the contributions without rest terms show negative values.*

And it would be nice to see an argument why not all reactions of the HOx and H chemistry are tagged. Computationally too expensive? In case all reactions were tagged, the rest terms would be needless, right?

*Yes indeed, if all species of the HO$_x$ and H chemistry would be tagged, then the rest terms would not be needed. However, there are restrictions for tagging all species: We suggest two implementations for tagging chemical species: (1) for short-lived species and (2) for long-lived species. Only few chemical species of the HO$_x$ and H chemistry can be tagged with the tagging implementation for short-lived species as only a few species are short-lived and are in steady-state. Therefore, it is necessary to exclude the reaction of H$_2$O$_2$ + hv, HOCl + hv and HOBr + hv (see Appendix A). The tagging implementation of the long-lived species could theoretically be applied to all species. However, each chemical species need to be split up in ten tagged categories. The total chemical mechanism in EMAC (MECCA) contains about 72 chemical species. Splitting up each chemical species into ten categories significantly increases the memory demand (tenfold increase). This is why we cannot tag all chemical species of the HO$_x$ and H chemistry and thus we introduce the rest terms. Consequently, the tagging mechanism presented here is a trade-off between accuracy and completeness as well as computation time and feasibility (see Chapter 6 Options, limitations, and future perspectives in Grewe et al. 2017).*

***Changes in manuscript:***
*However, this is not possible as the tagging method of short-lived species does not apply to all reactions of the HO$_x$ and H chemistry (for examples see Appendix A). Moreover, tagging all chemical species of the HO$_x$ and H chemistry with the implementation of long-lived species would significantly increase the memory demand of a climate simulation.*

- p16, l25: Replace "To quantify" with "To demonstrate"

   *As suggested by the reviewer, we replaced the word "quantify" with "demonstrate".*

- p16, l29/30: Doesn't a larger O3 contribution from shipping also lead to more "shipping" OH via the formation of O(1D) and the subsequent reaction with H2O?

   *In fact, the reaction rate of H$_2$O and O($^1$D) is about a magnitude smaller than the rates of the reactions of NO$_y$ and OH as well as the reactions of NO$_y$ and HO$_2$. Thus, the loss of HO$_x$ by NO$_y$ largely dominates over the production of HO$_x$ via the reaction of H$_2$O and O($^1$D).*

   ***Changes in manuscript:***
   *In this region NO$_y$ from shipping emissions is larger than in the Mediterranean Sea causing a reduction of OH and HO$_2$ by reactions (14) to (17).*

- references: Grewe et al., 2017 is already published and no longer a discussion paper.

   *Thank you for this hint. We updated the citation.*

- caption S1: firt -> first

*Thank you. We corrected it.*

**Response to Reviewer #2**

Many thanks to the authors for carefully revising their manuscript. I believe that these changes address all my concerns, and I am mostly happy with the manuscript as presented. I have two minor corrections that I would suggest are addressed prior to publication.

*Thank you very much for appreciating our efforts.*

1. Page 2 line 12: The tense is slightly incorrect in the sentence "On regional and global scale, no direct HOx measurement is available." - I would suggest perhaps "On regional and global scales, no direct HOx measurements are available."

*Thank you, we changed the text accordingly.*

***Changes in manuscript:***
*On regional and global scales, no direct HOx measurements are available.*

[revised manuscript text omitted]